# Developmental features and unique characteristics of peptide-specific PLZF⁺ innate-like T cells in mice

Ahmed Hassan [1], Nico Heise[1], Anja Schimrock[1], Stefanie Willenzon[1], Inga Ravens[1], Reinhold Förster [1,2,3] & Hristo Georgiev [1] ✉

Peptide-specific PLZF⁺ innate-like T (PILT) cells are a member of the innate-like T cell family utilizing a diverse set of T cell receptor (TCR) Vβ chains. Yet there are no present studies providing clues into the developmental features of PILT cells at a transcriptome level. Here, we performed single-cell transcriptomic analyses of PILT cells and compared them to other members of the innate-like T cell family. We show that PILT cells share similar transcriptional profiles and overlapping developmental trajectories with invariant Natural Killer T (iNKT) cells. However, in contrast to iNKT cells, PILT cells display a polyclonal TCR repertoire closely resembling the one of conventional CD8 T cells, inferring MHC I restriction and a broader range of antigen specificity. We further show that artificial thymic organoid cultures (ATOC) support selection and development of PILT cells in vitro exhibiting similar transcriptional profiles to their counterparts maturing in the thymus. Moreover, using an "on-time" TCR retrogenic ATOC system, we provide evidence for an instructive role of TCR specificity in PILT cell lineage commitment and functional differentiation. Altogether, our findings provide further insights into the PILT cells unique characteristics and molecular mechanisms governing their development.

The innate-like T cell lineage encompasses a diverse group of αβ and γδ T cells that exhibit features of both adaptive and innate immunity. Members of the innate-like T cell lineage acquire a distinct memory-like phenotype during development in the thymus allowing for a quick response to challenges in both T cell receptor (TCR) dependent and/or independent fashion[1,2]. Once challenged, innate-like T cells can respond in several ways including; (i) cytokine production[3,4] (ii) Fas-FasL mediated killing[5,6], and (iii) granzyme-perforin mediated killing[7]. Recent studies suggest that members of the innate-like T cell lineage play a role in modulating adaptive and innate responses for example by (i) supporting germinal center formation[8,9] (ii) enhancing cytotoxicity, interferon production, and proliferation of cytotoxic T cell[10], and (iii) modulating natural killer cells activation[11]. As prominent members of the αβ innate-like T cells, invariant natural killer T (iNKT)

cells, and mucosal-associated invariant T (MAIT) cells, are the most studied members of this cell family with a growing body of research painting a clearer picture of their intrathymic development[12–21]. During thymopoiesis, early thymic progenitors (ETP) give rise to all thymic T cells, this includes both innate-like and conventional T cells with both sharing the same developmental pathway at earlier stages but diverging at the double-positive (DP) stage. While double-positive (DP) thymocytes committed to conventional T cell lineage are selected by thymic epithelial cells, DP thymocytes committed to innate-like T cell lineage get selected by neighboring antigen-presenting DP thymocytes. At this stage, DP thymocytes committed to the Innate-like T cell lineage receive essential signals associated with the acquisition of the innate-like phenotype. These include a relatively strong TCR signaling (regarded as agonist selection)[22] and auxiliary co-receptor signaling,

[1]Institute of Immunology, Hannover Medical School, Hannover, Germany. [2]Cluster of Excellence RESIST (EXC 2155), Hannover Medical School, Hannover, Germany. [3]German Centre for Infection Research, Partner Site Hannover-Braunschweig, Hannover, Germany. ✉e-mail: georgiev.hristo@mh-hannover.de

via homotypic interaction between signaling lymphocytic activation molecules family (SLAMF) receptors SLAMF1 and SLAMF6, required for upregulation of the transcription factors EGR2[23] and PLZF[24]. Consequently, the upregulation of PLZF is reported to be responsible for the "innateness" of iNKT and MAIT cells[17,24] while modulating the transcription profile of selected thymocytes toward an innate-like phenotype[23,25].

We previously described peptide-specific PLZF+ innate-like T (PILT) cells[26] as a member of the innate-like T cell lineage. Similar to other members of the innate-like T cell lineage, PILT cells are found in three major functional subsets mirroring that of CD4 T helper and innate lymphoid cells[15,26–28]. The different subsets can be characterised based on expression of key transcription factors into PILT1 (PLZF^low, T-bet^hi, RORγt^neg), PILT2 (PLZF^hi, T-bet^neg, RORγt^neg), and PILT17 (PLZF^int, T-bet^neg, RORγt^hi)[26]. Akin to iNKT and MAIT cells, PILT cell development is SLAMF dependent and requires DP:DP thymocytes interaction, assuming that PILT cells follow a similar intrathymic developmental pathway as observed for iNKT and MAIT cells[26]. Yet unlike iNKT and MAIT cells, which recognise non-peptide antigens[29,30], PILT cells are considered to be selected on peptide antigens presented by a polymorphic major histocompatibility complex MHC-I or MHC-I-like molecules[26] and were shown to expand in a B6.CD4^Cre/Cd1d^-/-/Nlrc5-stop^flox mouse model, hereinafter referred to as "T-MHC I", due to conditional upregulation of classical MHC I expression in the DP stage of thymocyte development in this model.

In the present study, we investigate the transcriptional and the TCR repertoire landscapes of thymic PILT cells and compare it to iNKT cells using single-cell RNA sequencing (scRNAseq) coupled with single-cell V(D)J sequencing (scV(D)Jseq). Our results show that PILT cells are a heterogeneous population segregating into three major subsets that can be further grouped into several sub-clusters. Moreover, we provide evidence that PILT cells share transcriptional profiles with other members of the innate-like T cells lineage and we further validate these findings by using in house generated iNKT as well as publically available iNKT and MAIT cell datasets. In addition, an in-depth analysis of the scV(D)Jseq data suggests that in contrast to iNKT and MAIT cells, PILT cells exhibit a highly diverse TCR repertoire, indicating that PILT cells can respond to a broader range of antigens in comparison to other members of the innate-like T cells. Further, a comparative TCR analysis of complementarity-determining regions (CDRs) shows that PILT-derived CDR1/CDR2 regions share high similarity with conventional CD8 T cell TCR repertoires, inferring MHC-I restriction of PILT cells. Furthermore, we report that innate-like T cells develop in vitro in an artificial thymic organoid culture (ATOC) system exhibiting a similar transcriptional profile to their counterparts developing in the thymus. Lastly, we also show that PILT1 derived TCR clonotypes favor acquisition of PILT1 innate-like phenotype during T cell development in a T-cell receptor retrogenic ATOC system, suggesting that TCR specificity might have an instructive role in PILT cell lineage commitment and development.

## Results

### Transcriptional landscape of thymic PILT cell-enriched population

In analytical flow cytometry, PILT cells are identified by gating on TCRβ+PLZF+ T cell population after excluding other T cell subsets known to express PLZF, such as iNKT, MAIT and γδ T cells (Supplementary Fig. 1a)[26]. However, based on our current knowledge, PILT-cells lack unique surface markers distinguishing them from other innate-like and antigen-experienced T cells. Therefore, obtaining pure live PILT cell populations as required for scRNAseq is not possible right now. Interestingly, we noted that all PILT cells express high levels of either or both activation markers CD44 and PD1 (Supplementary Fig. 1a). Based on this, we devised a suitable sorting strategy excluding known innate-like T cells while enriching for antigen-experienced

T cells including PILT cells (Supplementary Fig. 1a). This sorting strategy was used to obtain thymic PILT cell-enriched populations from T-MHC I and Cd1d^-/- mice. The latter strain serving as a suitable model for analyzing WT PILT cells due to complete abrogation of NKT-cell development and therefore avoiding possible contamination of sorted PILT cell-enriched fractions with CD1d restricted NKT cells. In parallel, thymic iNKT cells from B6 mice were sorted for further analyses (Supplementary Fig. 1a).

In total, three independent experiments were performed as depicted in Supplementary Fig. 1b. In the first part of the analysis, we extracted the PILT enriched population captured in GEM1:4 (excluding iNKT cells) based on hashtag signal. We also excluded low-quality cells, doublets, cells with no or missing TCR information and cells with TRAV1-TRAJ33 chains identified as MAIT cell contamination. In total 26148 cells from PILT cell-enriched population sorts were visualised on a two-dimensional Uniform Manifold Approximation and Projection (UMAP) (Fig. 1a). The cells were spread into 14 distinct clusters representing a mix of multiple thymocyte development stages and several effector T cell subsets. All clusters were annotated using data-driven differentially expressed genes (DEGs) and publicly available curated markers (Fig. 1a, c). DP thymocytes (DP) in cluster 0 co-expressed Cd8b1 and Cd4 and had high expression of genes associated with VDJ recombination including Arpp21[31], Dntt, and Rag1[32] (Fig. 1b, c). Thymocytes undergoing positive selection (Tsel) in cluster 1 upregulated genes associated with TCR signaling (Cd5[33], Satb1[34] and Cd69[35]), intrathymic migration (Ccr4[36], Ccr7[37] and Ccr9[38]) and immune synapse formation (Cd2[39]) (Fig. 1b, c). Both committed CD4 single-positive thymocytes (CD4 SP) and CD8 single-positive thymocytes (CD8 SP) in clusters 2 and 3, respectively, were enriched for expression of Foxo1, Ccr7, Sell, S1pr1, and Klf2 (Figs. 1b, c). This suggests that both clusters represent thymocytes that have reached the end of intrathymic development and were ready to egress from the thymus to the periphery[40]. Circulating Treg (cTreg) cells in cluster 4 had high expression of Foxp3, Ccr6, Nt5e, and Il18r1 all in line with previous reports describing a population of Tregs that re-enter the thymus from the periphery and accumulate by age (Fig. 1b, c)[41–43]. T cells in cluster 5 expressed Cd4 while upregulating that of Ly6a, Il7r, Itgb1, Tbx21 (encoding T-bet), Ccl5 and Cxcr3, resembling a CD4 effector memory (CD4 T_EM) phenotype with TH1 polarisation with several studies reporting instances of such populations re-entering the thymus (Fig. 1b, c)[44–47]. CD8+ Memory-like T cells (T_ML) in cluster 6 expressed Ccl5, Ly6c2, Cxcr3, Eomes, Sell and Il2rb, all genes associated with memory-like phenotype[48,49]. Cells in cluster 7 exhibited high expression levels of Cd5, Nr4a1 and Ikzf2 (the latter two encoding Nur77, and Helios, respectively) indicating a transcription profile similar to that of agonist-signaled thymocytes (T(agonist)) (Fig. 1b, c)[50]. Cells in cluster 8 displayed low expression levels of Cd8b1, Cd4 and Il2ra. However, they showed relatively high expression of Id3, Pdcd1 (encoding PD-1), Trac, and Cd5, respectively, an observation in line with previous reports characterising thymic intraepithelial lymphocyte progenitors (IEL_p) (Fig. 1b, c)[51,52]. Cycling thymocytes in cluster 9 differentially expressed Mki67, Hist1h2ae, Top2a, and Ube2c indicating an active cell cycle (Fig. 1b). γδ24 T cells in cluster 10 were characterised by high Trdc, Sox4, Gzma, Blk and Rorc expression[53] (Fig. 1b).

Lastly, clusters 11, 12 and 13 were comprised of PILT cells with upregulated expression of Zbtb16 (encoding PLZF), albeit at a different level, where each of the three PILT cell clusters represents a unique functional subset similar to that reported for iNKT and MAIT cells[15,27,54,55]. Akin to iNKT1 cells, PILT1 cells in cluster 11 upregulated a combination of NK and activation markers including Nkg7, Klrk1, Ifng, Ccl5, Il2rb and Tbx21 (Fig. 1b, c). PILT2 cells in cluster 12 exhibit the highest Zbtb16 and Il4 expression (Fig. 1b, c). Expression of Il1r1, Il23r, Blk, Serpinb1a and, Rorc (encoding RORγt) (Fig. 1b, c) was upregulated in PILT17 cluster indicating a transcriptional profile similar to that of the iNKT17 subset. Therefore, these results indicate that

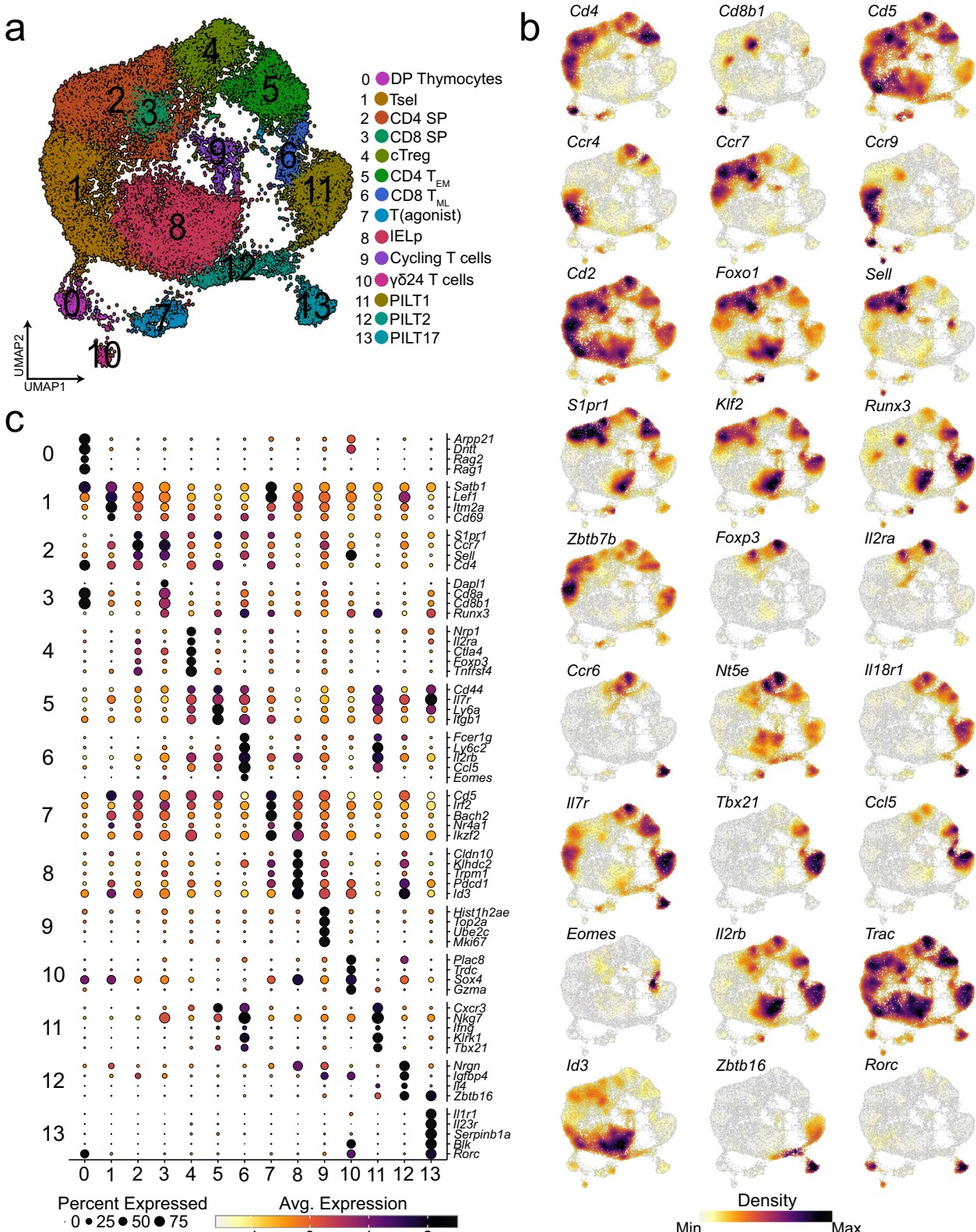

**Fig. 1 | Transcriptional landscape of sorted thymic PILT cell-enriched population. a** UMAP of sorted thymic PILT cell-enriched population from T-MHC and Cd1d$^{-/-}$ mice after QC. Displayed are 26148 cells derived from three biologically independent experiments as depicted in Supplementary Fig. 1b. DP is an abbreviation for double positive thymocytes; Tsel is for thymocytes undergoing positive selection; CD4 SP for CD4 single positive thymocytes; CD8 SP for CD8 single positive thymocytes; cTreg for circulating regulatory T cells; CD4 T$_{EM}$ for CD4 effector memory T cells; CD8 T$_{ML}$ for CD8$^+$ Memory-like T cells; T(agonist) for agonist-signaled thymocytes; IELp for intraepithelial lymphocyte progenitors; γδ24 T cells for CD24$^+$ gamma-delta T cells; and PILT for peptide-specific PLZF$^+$ innate-like T cells. **b** Dot plot showing the scaled log normalised average expression of selected differentially expressed genes (DEGs) and publicly available curated markers for each of the clusters depicted in (**a**). **c** Density plots of selected genes.

all major functional PILT subsets are present in the generated datasets for further analysis. Overall, we generated a dataset capturing the transcriptional landscape of conventional and unconventional T cells in the thymus including; T(agonist), IELp, CD8 $T_{ML}$, cTregs, CD4 $T_{EM}$ and most importantly the different functional subsets of PILT cells.

## PILT cells further subdivide into unique functional subsets similar to that of iNKT cells

Recent studies utilizing scRNAseq approach reported further heterogeneity within thymic iNKT cell subsets[14,16,18,20] beyond the well-known classification into iNKT1, 2 and 17 cells. Therefore, we inquired for a so far hidden diversity within the PILT cell fraction by a separate cluster-analysis of the 3995 PILT cells present in clusters 11, 12 and 13 as shown in Fig. 1a. This yielded eight clusters (Fig. 2a) which were then analysed side by side with sorted iNKT cells. In total 5403 iNKT cells were found to distribute into six clusters (Fig. 2c). Using curated gene markers we identified the same four major subsets (subsets 0, 1, 2, and 17) in both PILT and iNKT cell populations (Fig. 2a–d). PILT0 and iNKT0 subsets were present in single clusters exhibiting high expression of *Id3*, *Nrgn*, and *Lef1*, reported to be upregulated in earlier stages in innate-like T cell development[14] (Fig. 2b, d). Furthermore, gene set enrichment analysis (GSEA) (Supplementary Fig. 2a) shows an enrichment for genes associated with earlier stages in T cell development in these clusters, inferring that both PILT0 and iNKT0 cell clusters represent an early stage of PILT and iNKT cell development. Expectedly, PILT2 and iNKT2 subsets displayed the highest expression of *Zbtb16* and an upregulated expression of *Tesc* (coding for a calcium-binding protein)[56], *Il4*, *Plac8* and *Il6ra* (Fig. 2b, d and Supplementary Fig. 2b, c). Interestingly, the PILT2 cell subset segregated into two distinct clusters (PILT clusters 1 and 2). While both clusters share similarities, cluster 1 had higher expression of *Ccr7* (required for intrathymic migration)[37] whilst NK-related genes (such as *Klra1* and *Nkg7*) were upregulated in PILT cluster 2 (Fig. 2b and Supplementary Fig. 2b–f). This suggests that PILT cluster 2 encompasses PILT cells committed to the PILT1 lineage, whereas cluster 1 harbors a CCR7⁺ PILT cell population reminiscent of the CCR7⁺ multi-potent iNKT progenitors previously reported by Wang et al.[57]. However, we could not reliably reproduce these findings in our iNKT cells dataset where the iNKT2 subset consisted only of cluster 1, most likely due to the low frequency of iNKT2 cells in our iNKT sorts (Fig. 2c, d). PILT1 and iNKT1 subsets defined by their high expression levels of *Tbx21*, *Nkg7*, *Ccl5* and *Il2rb* (Supplementary Fig. 2b, d) showed further heterogeneity within by further segregating into four PILT clusters 3–6 and three iNKT clusters 2–4 (Fig. 2a–d). PILT cluster 3 and iNKT cluster 2 displayed upregulated expression of multiple Interferon-Stimulated Genes (ISG) (Fig. 2b, d). PILT cluster 4 and iNKT cluster 3 resemble recently committed PILT1 and iNKT1 as both still, albeit at a lower level, express genes associated with PILT2 and iNKT2 characteristics such as *Izumo1r*, *Il4*, and *Icos* (Supplementary Fig. 2b, e) while acquiring the gene signature associated with PILT1 and iNKT1 phenotype. Moreover, both clusters displayed low expression levels of *Klf2* and *S1pr1* (Supplementary Fig. 2b, e) indicating that both still need to undergo further maturation steps before they are licensed to egress from the thymus. In contrast, PILT clusters 5, 6, and iNKT cluster 4 represent bona fide PILT1 and iNKT1 cells ready to egress from the thymus to the periphery as they upregulated *Klf2* and *S1pr1* (Supplementary Fig. 2b, e) while also expressing NK-related markers (*Nkg7*, *Klra7*, and *Klra9*) and genes mediating cytotoxicity (*Prf1*, *Gzmb*, *Gzma*, *Faslg* and *Tnsfs10*) (Supplementary Fig. 2b, f). Lastly, transcripts of *Tmem176a*, *Serpinb1a*, *Ckb*, *Blk* and *Rorc* (Fig. 2b, d and Supplementary Fig. 2b) were enriched in both PILT17 and iNKT17 subsets. Module scores of previously reported iNKT0, iNKT1, iNKT2 and iNKT17 markers (Supplementary Table 1)[28,54,55] was in line with our annotation further validating our findings in both PILT and iNKT populations (Fig. 2e, f).

In a recent study, Baranek et al. defined four subpopulations of iNKT1 cells (iNKT1-ISG, iNKT1a, iNKT1b, and iNKT1c) and two

subpopulations of iNKT2 cells (iNKT2a and iNKT2b) based on transcriptional heterogeneity found within these two iNKT subsets[16]. Therefore, we asked whether these sub-populations correspond to the PILT clusters identified above. To this end, we calculated the module scores of the top 50 DEGs unique to each subpopulation as reported by Baranek et al. (Fig. 2g). iNKT0, iNKT1-ISG and iNKT17 gene expression signatures were enriched in clusters 0, 3 and 7, respectively. High enrichment of iNKT1a gene expression signature was present in clusters 2 and 7. An iNKT1b-like expression signature was enriched in clusters 4 and 6, the former representing a cluster committed to PILT1 subsets, yet not fully differentiated as mentioned above. Genes associated with iNKT1c transcriptional signature were enriched in clusters 5 and 6 further supporting the notion that these clusters represent fully differentiated PILT1 cells. Lastly, the module score of iNKT2a and iNKT2b expression signature was up in several clusters with the highest score of an iNKT2a-like signature in cluster 1 and the highest score of an iNKT2b-like signature in cluster 2 (Fig. 2g). Taken together, these data suggest that PILT-cluster 1 resembles iNKT2a cells, PILT-cluster 2 exhibits a mixed signature of iNKT2b/iNKT1a cells, PILT-cluster 4 resembles iNKT1b cells and PILT-cluster 5 is similar to iNKT1c cells while cells in cluster 6 display a mixt signature of iNKT1b/c cells (Fig. 2h).

In silico development trajectory built using Monocle3 (Fig. 2i) indicates that PILT cells follow a similar developmental trajectory as reported before for iNKT cells[16]. This analysis suggested a forked trajectory starting from the PILT0 cluster and branching later in the PILT2a cluster, which had the highest expression of *Ccr7*. These results are in line with the previous report by Baranek et al.[16], suggesting by analogy to iNKT cells that the PILT2a cluster harbors CCR7⁺ multi-potent PILT cell progenitors capable of further differentiating into effector subsets in the thymus. At the PILT2a cluster, the trajectory bifurcates into either PILT1 or PILT17 direction with the PILT1 trajectory splitting up later on again suggesting a non-linear development model for PILT1 cells. At the start of the trajectory, iNKT0 gene signature score was the highest and as the trajectory advanced this score started to gradually decline while iNKT1 and iNKT17 gene signature scores reached their peaks at the respective ends of the trajectory (Fig. 2j).

## PILT cells share a similar transcriptional profile with other innate-like T cells

PILT and iNKT cells are heterogeneous cell populations yet are comprised of similar subsets. In addition, the in silico analyses indicate that both share comparable developmental trajectories. To further investigate putative similarities between both populations we integrated both, PILT (Fig. 2a), and iNKT cells (Fig. 2c) datasets into one dataset for further analysis. In total, 9398 cells spread across eight clusters (Fig. 3a). The four major subsets were identified based on module scores of previously reported iNKT subset markers (Supplementary Fig. 3a and Supplementary Table 1) while each cluster was annotated based on data-driven DEGs analysis (Supplementary Fig. 3b). iNKT and PILT cells were represented in all clusters albeit not in equal ratios with most notable differences in cluster 0 (PILT0/iNKT0) and cluster 6 (PILT17/iNKT17) where most of the cells were of PILT cell origin (Fig. 3b).

Next, we inquired for distribution of PILT cells derived from T-MHC I mice hereinafter (referred to as "MHC-PILT") and PILT cells derived from CD1d⁻/⁻ mice (referred to as "WT-PILT") and iNKT cells. iNKT, MHC-PILT and WT-PILT cells were distributed across all clusters except for cluster 7 representing recently selected thymocytes exiting DP stage of iNKT and WT-PILT cells origin (Fig. 3c left panel). Expectedly, the vast proportion of iNKT cells had iNKT1 phenotype with the rest having iNKT0, iNKT2 or iNKT17 phenotype (Fig. 3c right panel). MHC and WT-PILT cells exhibited a similar distribution pattern albeit not to the same extent with higher PILT0, PILT2 and PILT17 frequencies

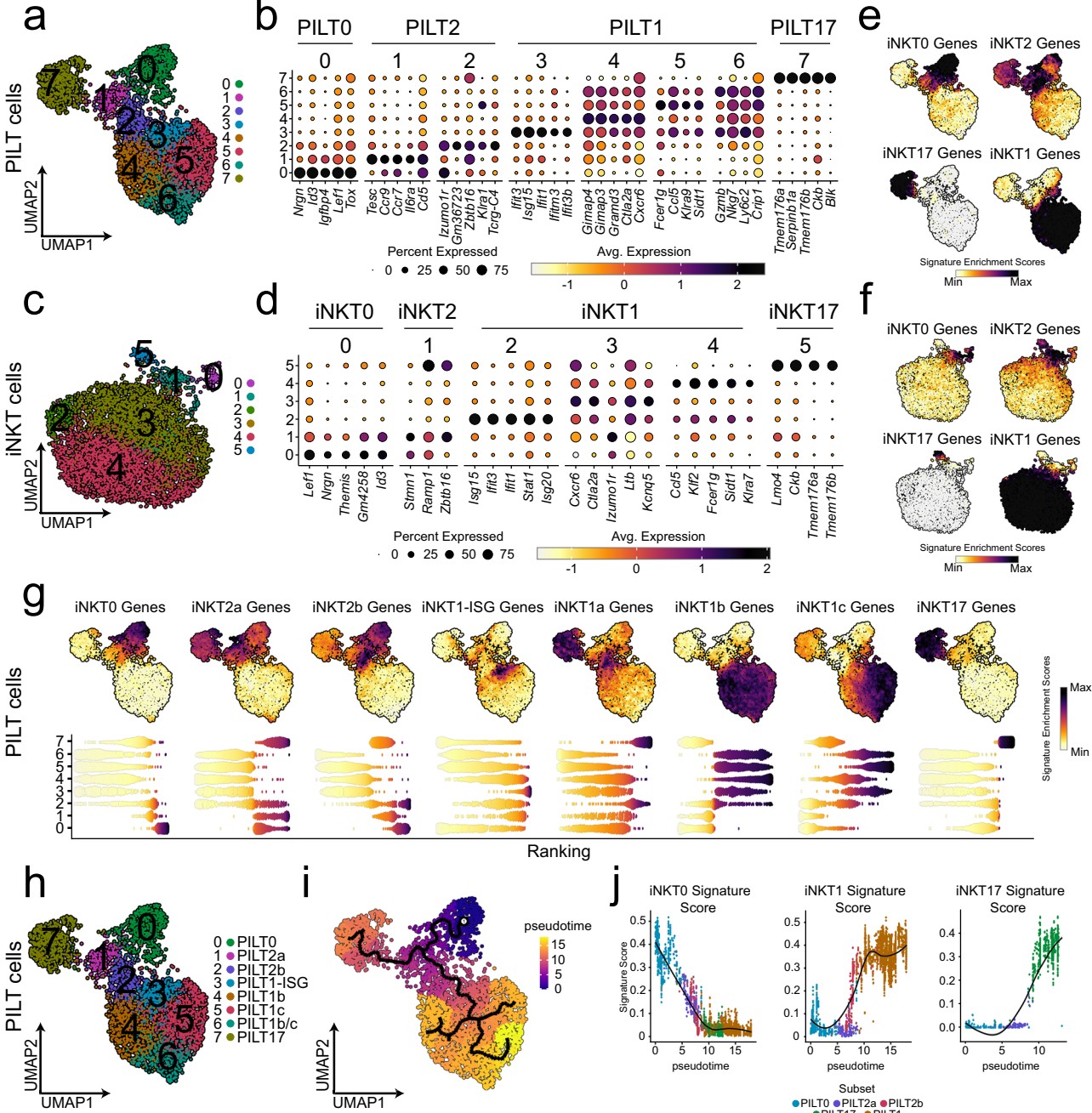

**Fig. 2 | Transcriptional landscape of PILT and iNKT cells. a** UMAP of subsetted PILT cells (cells from clusters 11, 12 and 13 from Fig. 1a) *n* = 3995 cells. **b** Dot plot showing the scaled log normalised average expression of the top 5 differentially expressed gene markers for each PILT cell cluster. **c** UMAP of sorted iNKT cells *n* = 5403 cells derived from two biologically independent experiments as depicted in Supplementary Fig. 1b. **d** Dot plot showing the scaled log normalised average expression of the top 5 differentially expressed gene markers for each iNKT cell cluster. Feature plot showing the k-nearest neighbors (KNN) smoothed UCell signature enrichment score for iNKT subsets signature gene markers (Supplementary

Table 1) for PILT cells in (**e**) and iNKT cells in (**f**). **g** Feature (top) and bee swarm (bottom) plots for KNN smoothed UCell signature enrichment score of the top 50 iNKT subsets gene markers reported by Baranek et al.[16] for PILT cells. **h** UMAP of subsetted PILT cells showing the annotiation of each cluster. **i** UMAP showing PILT cell pseudo-time trajectory built using Monocle3. The cells are ranked in pseudo-time based on *Egr1, Egr2, Cd24a, Rag1* and *Rag2* expression. **j** Gene signature score dynamics in pseudotime. PILT is an abbreviation for peptide-specific PLZF[+] innate-like T cells; iNKT for invariant natural killer T cells and ISG for interferon-stimulated genes.

(Fig. 3c right panel). Pearson's correlation assay (Supplementary Fig. 3c) suggests that regardless of its source, different cell types within the same cluster correlate more with each other than they correlate with the same cells type from different clusters. Moreover, within the same cluster, all cell types had similar expression levels of the top DEGs defining each cluster (Supplementary Fig. 3d), proving that the cell type does not influence how the cells cluster.

These findings were further validated by integrating our PILT cell dataset with publicly available datasets of iNKT cells from Baranek et al.[16] and MAIT cells from Legoux et al.[15] PILT cells overlapped with Baranek's iNKT cells in all clusters albeit with different ratios (Fig. 3d, e). PILT cells also overlapped with Legoux's MAIT cells in all clusters however MAIT cells were underrepresented in cluster 2 which represented PILT1 and MAIT1 due to the low numbers of captured

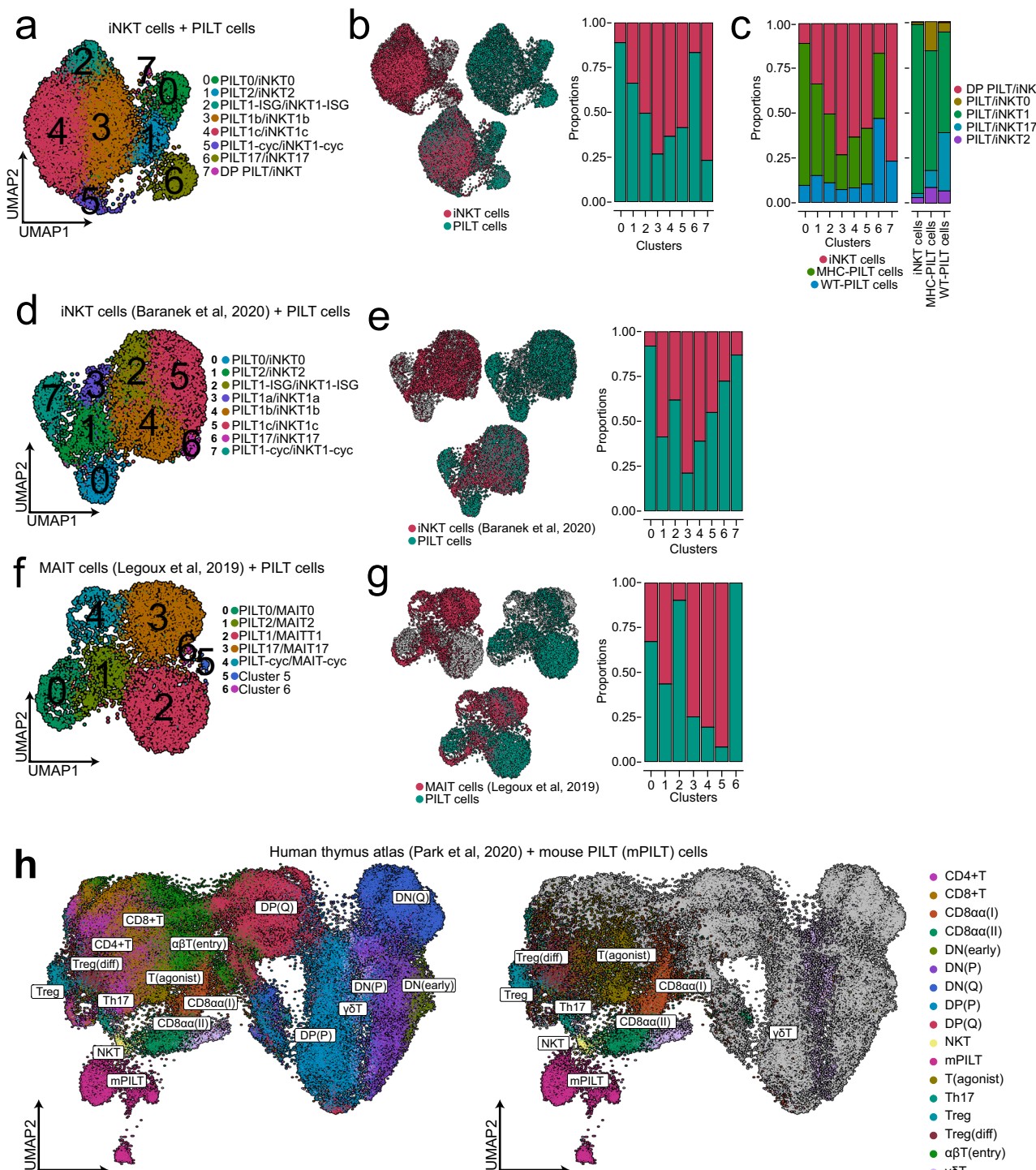

MAIT1 cells in the original paper (Fig. 3f, g). Markedly, when integrated with an atlas of human thymic development dataset[58], mouse PILT cells from our dataset clustered closely with human NKT-like cells and other unconventional populations with innate-like phenotype highlighted in Fig. 3h. Collectively, these data demonstrate that PILT cells share similar transcription signature and development trajectory with other members of the innate-like T cell lineage such as iNKT and MAIT cells.

**Unlike iNKT cells, PILT cells possess a diverse TCR repertoire**

Next, we assessed the diversity of the TCR repertoire present on PILT cells by analyzing the PILT and iNKT cell scV(D)Jseq dataset generated in this study, in parallel with PBMC datasets from B6

mouse published by 10x Genomics[59]. Simpson diversity index showed that PILT cells had a TCR repertoire diversity comparable to that of thymic and peripheral CD8 and CD4 T cells thus being far more complex than that of iNKT cells (Fig. 4a). TRAV9N-3 was the top TCRα Vgene used by PILT cells (15.1% in WT-PILT cells and 7.9% in MHC-PILT cells) while TRBV13-3 was the most used TCRβ Vgene (WT-PILT cells 14.5% and MHC-PILT cells 10%) (Fig. 4b). TRAV9N-3 seem to favor pairing with TRBV16 (4.6% in WT-PILT and 1.7% in MHC-PILT) much more than pairing with TRBV13-3 (1.1% in WT-PILT and 0.74% in MHC-PILT) (Fig. 4c). This all suggests that PILT cells are not restricted to a specific antigen but rather recognize a broad range of antigens.

**Fig. 3 | PILT cells share a similar transcriptional profile with other innate-like T cells. a** UMAP of integrated PILT (from Fig. 2a) and iNKT cells (from Fig. 2b) with $n = 9398$ cells. **b** Distribution of iNKT and PILT cells shown as UMAP overlay (left panel). Right panel, filled bar plot displaying iNKT and PILT-cell distribution across the different clusters as annotated in (**a**). **c** Filled bar plots showing the distribution of iNKT, MHC-PILT and WT-PILT cells in each cluster (left panel). A filled bar plot showing the subset composition of iNKT, MHC-PILT and WT-PILT cells (right panel). WT-PILT is an abbreviation for derived from CD1d$^{-/-}$ mice and MHC-PILT for derived from T-MHC I mice peptide-specific PLZF$^+$ innate-like T. **d** UMAP of integrated PILT cells (from Fig. 2a) and iNKT cells from Baranek et al.[16] $n = 7226$ cells. **e** Distribution of iNKT (Baranek et al.) and PILT cells shown as UMAP overlay (left panel). Right panel, filled bar plot displaying iNKT and PILT-cell distribution across the different clusters as annotated in (**d**). **f** UMAP of integrated PILT cells (from Fig. 2a) and MAIT cells from Legoux et al.[15]. $n = 7339$ cells. **g** Distribution of MAIT

(Legoux et al.) and PILT cells shown as UMAP overlay (left panel). Right panel, filled bar plot displaying iNKT and PILT-cell distribution across the different clusters as annotated in (**f**). **h** UMAP of integrated PILT cells (from Fig. 2a) and an atlas of human thymic development dataset with thymocyte-cell populations as annotated by Park et al. $n = 80989$ (left UMAP). Unconventional thymocyte-cell populations are highlighted in the second UMAP. CD8αα is an abbreviation for CD8 alpha-alpha; DN(early) for double negative early thymic progenitor; DN(P) for double negative proliferating; DN(Q) for double negative quiescent; DP(P) for double positive proliferating; DP(Q) for double positive quiescent; NKT for natural killer T-like cells; mPILT for mouse peptide-specific PLZF$^+$ innate-like T; T(agonist) for agonist-signaled; Th17 for T helper 17-like cells; Treg for regulatory T cells; Treg(diff) for differentiating regulatory T cells; αβT(entry) for alpha-beta T cells entering single positive stage and γδT for gamma-delta T cells. Source Data are provided as a Source Data file.

It was previously shown that CDR1 and CDR2 TCR regions serve as a point of contact between the TCR and the conserved α-helices of the MHC molecules[60,61]. Therefore, we sought to determine whether PILT derived CDR1/CDR2 TCR regions resembled CD4 or CD8 T cell repertoire. Indeed, Horn-Morisita index score analysis showed that MHC-PILT and WT-PILT cells shared higher degree of CDR1/CDR2 similarities −values to 0− with CD8 T cells when compared to CD4 T cells (Fig. 4d). Moreover, PILT cells exhibit a Vgene-α and Vgene-β usage pattern which resemble CD8 T cells rather than CD4 T cells (Supplementary Fig. 4a, b).

Considering that PILT-cell development in the thymus is abrogated in B2m$^{-/-}$ animals (Supplementary Fig. 5), these data further corroborate that PILT cells recognize peptide antigens presented by MHC-I or MHC-I like molecules.

## Artificial thymic organoids support the development of conventional and unconventional T cells

Artificial thymic organoid cultures (ATOCs) represent a valuable tool for studying thymopoiesis in in vitro settings. However, reproducing thymic PILT cell development in vitro requires establishing a suitable ATOC system that can support not only selection and development of conventional but also that of innate-like T cells. Using an adapted version of Montel-Hagen et al.[62] protocol, where a modified mouse MS5-mDLL1/DLL4 stromal cell line provides Notch signaling (Supplementary Fig. 6a and Supplementary Fig. 7a), we tracked the development of DN thymocytes in this ATOC system weekly over the span of five weeks. Flow cytometry analysis (Fig. 5a) shows that most TCR$^-$ cells in the ATOC start differentiating into either γδ T cells or αβ T cells over five weeks reaching the highest percentage at week 3 and plateauing over the rest of the weeks. Gating on TCRγδ$^-$ cells we observed the dynamics of thymocyte development (Fig. 5b and Supplementary Fig. 7c) starting from the immature DN (CD4$^-$CD8$^-$TCR$^-$) passing through ISP (CD4$^-$CD8$^+$TCRαβ$^{lo}$) to the DP stage (CD4$^+$CD8$^+$) and finally maturing into either mature CD8 T cells (CD8$^+$TCRαβ$^{hi}$) or mature DN T cells (CD8$^-$CD4$^-$TCRαβ$^{hi}$). Of note, CD4 T cells did not develop since in this system MHC-II is not expressed on antigen presenting cells. When focusing on PILT cell development, a noticeable fraction of TCRβ$^+$PLZF$^+$ cells started emerging as early as weeks 1 and 2 reaching the highest frequency at weeks 3 and 4 (Fig. 5c). Notably, TCRβ$^+$PLZF$^+$ cells at weeks 1 and 2 had low CD44 expression whereas most of the TCRβ$^+$PLZF$^+$ cells at weeks 3 and 4 had already upregulated CD44 expression with an emergence of a prominent CD44$^+$NK1.1$^+$ cell fraction (Fig. 5d). By week 5 the ATOC system becomes overgrown by the MS5 stromal cells and reaches a point where the supplemented media is no longer sufficient to provide a suitable level of nutrient/cytokines required to sustain the system resulting in dramatic decrease in cell viability (Fig. 5e).

To gain further insight into the T cell development in the ATOC system, we sorted DN thymocytes (Supplementary Fig. 7b) used to seed ATOC and total live thymocytes collected from ATOCs at

different time points and used the cells to perform scRNAseq (Supplementary Fig. 1c). In the subsequent analyses, we separated γδ and αβ T cells (using Totalseq TCR γ/δ antibody). A total of 6884 TCR γ/δ thymocytes were found to split into 15 clusters representing different ATOC developmental stages (Fig. 6a). The generated clusters were annotated based on data-driven DEGs and a selection of curated markers (Fig. 6d, e and Supplementary Fig. 8a). At week 0, thymocytes were primarily at DN1-3a stage (cluster 0) and some cells were past β-selection entering DN3b/4 stage (cluster 1). By week 1, most of the thymocytes reached DP stage (cluster 4) and at week 2 a cluster of recently selected T cells emerged (cluster 6). Organoids at weeks 3, 4 and 5 were populated with mature SP thymocytes exhibiting naïve or effector memory phenotype (clusters 7-11) (Fig. 6a, b). Of note, DP thymocytes in cluster 4 upregulated expression of *Cd1d1*, *Sh2d1a* (encoding SAP), *Slamf1*, and *Slamf6* (Fig. 6c) required for innate-like selection/development at this stage[25].

Next, we sorted out iNKT cells and PILT cells enriched population from different ATOC representing different time points according to the gating strategy used previously for mouse thymi (Supplementary Fig. 1a, c). A total of 14,019 cells were found to split into 11 clusters (Fig. 7a), further annotated based on data-driven DEGs (Fig. 7b, c). Markedly, cluster 10 encompassed innate-like T cells with upregulated *Zbtb16* expression (Fig. 7d). In a following step, we sub-clustered cells from cluster 10 alongside all *Zbtb16* expressing cells (*Zbtb16* expression count >0). A total of 173 innate-like T cells, including PILT and iNKT cells (cells with TRAV11-TRAJ18 chains), distributed into four clusters (Fig. 7e) annotated based on selection of DEGs and curated selected markers (Fig. 7g–i).

Most of iNKT cells were derived from week 2 organoid and exhibited an iNKT2/17 mix phenotype (cluster 1) while PILT cells comprised the majority of innate-like T cells from weeks 3, 4 and 5 organoids (Fig. 7f) displaying a progenitor (cluster 0) or PILT1 phenotype (clusters 2 and 3). Interestingly, although ATOC-generated PILT cells seem to have a diverse TCR repertoire comparable to that of thymus-generated PILT cells (Supplementary Fig. 8b), most of the PILT cells followed the PILT1 developmental trajectory suggesting that this ATOC system favors PILT1 development.

## Artificial thymic organoids support the development of T-cell receptor retrogenic PILT1 cells

Next, we sought to investigate if defined TCR specificity can commit developing thymocytes to the innate-like T cell developmental pathway in the ATOC system and if the TCR specificity plays a role in shaping an effector phenotype. Of note, early TCRα expression can lead to premature thymocyte deletion during selection[63]. Therefore, we utilized a retroviral TCR delivery vector allowing "on-time" expression of TCRα in B6.Rag2$^{-/-}$CD4$^{Cre}$ settings together with continuous expression of TCRα and the congenic marker Thy1.1 (Fig. 8a and Supplementary Fig. 6b). Initially, we assessed this retrogenic ATOC system with two TCRαβ clonotype sequences with known specificities

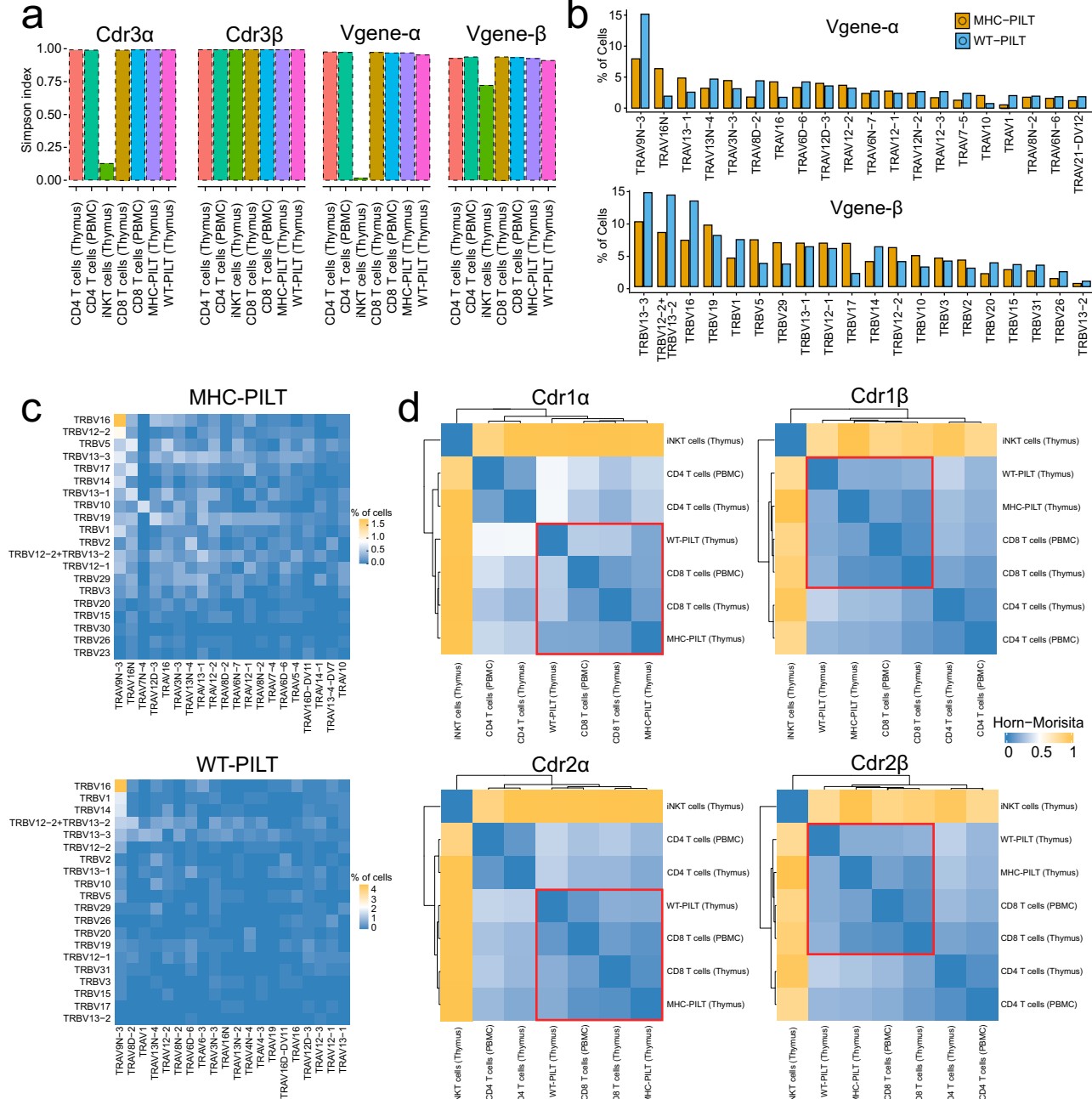

**Fig. 4 | Characteristics of PILT cell TCR repertoire. a** A bar plot showing the Simpson diversity index score of Cdr3α, Cdr3β, Vgene-α and, Vgene-β usage for down sampled PILT cells and other selected populations from the thymus (this experiment) and PBMCs (10x genomics published datasets). **b** A bar plot showing the Vgene-α and Vgene-β usage for MHC-PILT and WT-PILT cells. **c** A Heatmap shows the Vgene-α and Vgene-β pairing of MHC-PILT and WT-PILT cells. **d** A

heatmap showing the Horn-Morisita index score of Cdr1α, Cdr1β, Cdr2α and Cdr2β for PILT cells and other selected populations from the thymus (this experiment) and PBMCs (10x genomics published datasets). WT-PILT is an abbreviation for derived from CD1d−/− mice; MHC-PILT for derived from T-MHC I mice peptide-specific PLZF+ innate-like T; iNKT for invariant natural killer T cells and PBMC for peripheral blood mononuclear cells. Source Data are provided as a Source Data file.

(the OTI and an iNKT1 TCRαβ clonotype sequences, the latter extracted from our dataset). Flow cytometry analysis of the ATOCs after four weeks revealed that most of the developed thymocytes expressed Thy1.1 and had surface expression of either OTI or iNKT TCRs identified by their respective tetramer staining (Fig. 8b). Expectedly, the tetramer+ population from the OTI-ATOC developed as CD8 SP thymocytes while those from iNKT-ATOC were either CD8 SP or DN at this time point (Fig. 8b). Notably, although a small fraction of CD44hi cells developed in both groups, only the CD44hi cells from the iNKT-ATOC acquired an innate-like phenotype with upregulated expression of NK1.1, PLZF and T-bet (Fig. 8b). Therefore, these data suggest that

developing thymocytes bearing a TCRαβ clonotype with an innate-like T cell specificity can develop and acquire an innate-like phenotype in the ATOC system, although at low frequency.

The MS5 stromal cell line is originally derived from C3H/He mouse strain thus possessing an H-2k haplotype. Since our datasets are generated on B6 background, in a following step, we transduced the MS5-mDLL1/DLL4 cell line in order to express H-2Kb and H-2Db (Supplementary Fig. 7a). Next, to investigate for PILT cell development in the retrogenic ATOC system, we selected six PILT TCR clonotype sequences from our dataset (three PILT1 and three PILT17) (Fig. 8c) which were tested in a setup of retrogenic ATOCs as depicted in

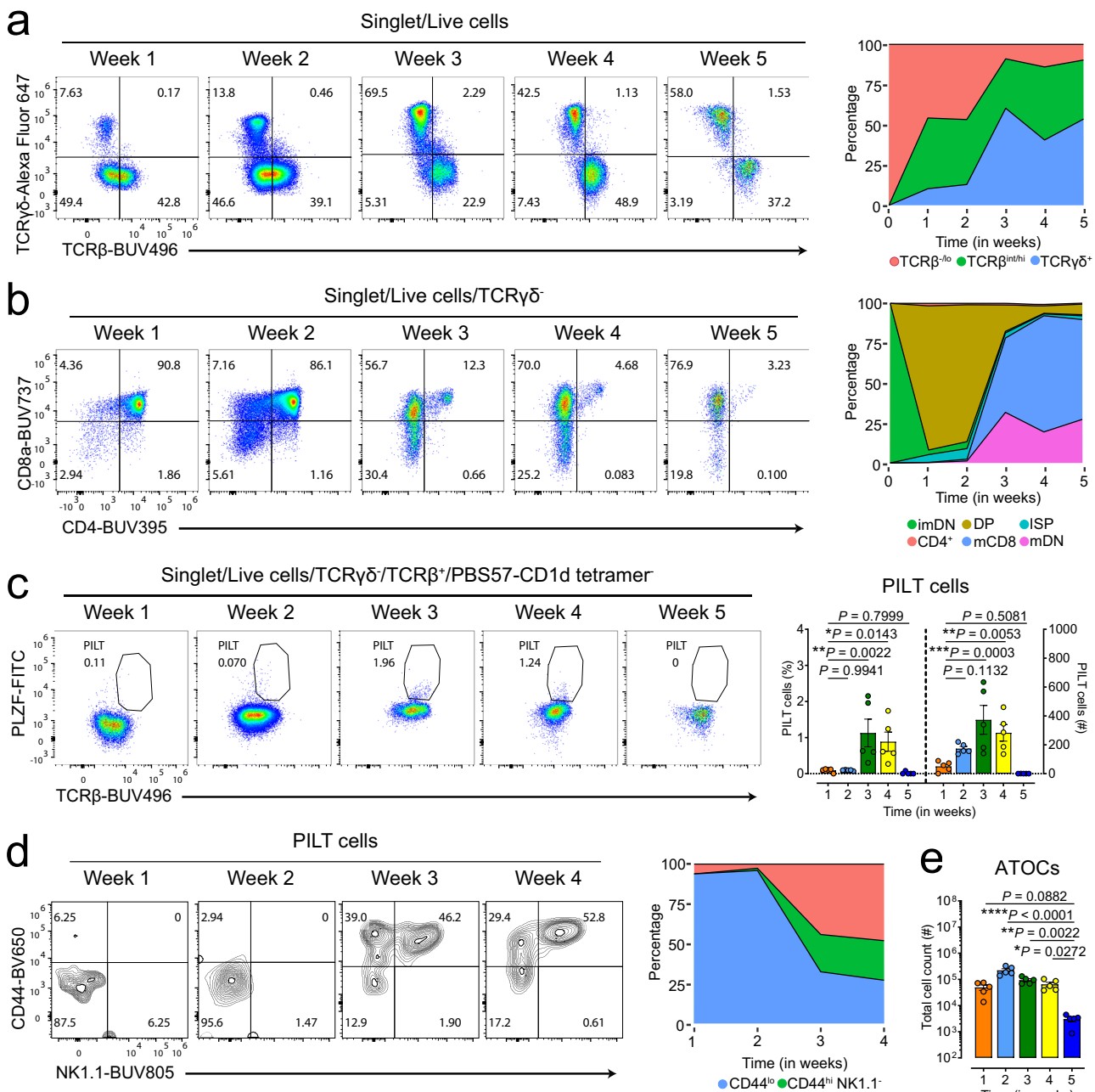

**Fig. 5 | Kinetics of DN thymocyte development in ATOC system over five weeks.**
**a** Representative flow cytometry plots (left) and a stacked area plot (right) showing the dynamics of TCRγδ+ and TCRαβ+ cells development in the ATOC system.
**b** Representative flow cytometry plots (left) and a stacked area plot (right) showing the dynamics of CD8+ and CD4+ T cells development in the ATOC system over time. DP is an abbreviation for double positive; imDN is for immature double negative; ISP for immature single positive; CD4+ for CD4 single positive; mCD8 for mature CD8 single positive and mDN for mature double negative T cells. **c** Representative flow cytometry plots (left) and a bar plot (right) showing the dynamics of PILT cells development in the ATOC system. **d** Representative flow cytometry plots (left) and

a bar plot (right) showing the dynamics of CD44 and NK1.1 expression on developing PILT cells in the ATOC system. Cells were gated as shown in Supplementary Fig. 7c. **e** Total cell counts of thymocytes developing in the ATOC system over time. Each point represents one ATOC in (**c**, **e**): $n = 4$ ATOCs per time point (**a**–**e**). Data are representative of two biologically independent experiments (**a**–**e**). Statistical significance was calculated using one-way ANOVA followed by Fisher's LSD multiple comparisons test; not significant ($P \geq 0.05$), *$P < 0.05$, **$P < 0.01$, ***$P < 0.001$ and ****$P < 0.0001$. Data are presented as mean values ± SEM. Source data are provided as a Source Data file.

Supplementary Fig. 6c. Flow cytometry analysis at week 4 showed that the retrogenic ATOCs seeded with PILT1 clonotypes developed a sizable CD44hi population whereas the corresponding PILT17 clonotypes were comparable in size to the control OTI TCR (Fig. 8d, e). Moreover, the fraction of CD44hi cells in two out of the three tested PILT1 clonotypes acquired relatively higher PLZF and T-bet expression in comparison to the small fraction of CD44hi cells detectable in the

control OTI TCR (Fig. 8f, g). Furthermore, most of the CD44hi retrogenic PILT cells downregulated CD4 and CD8α expression during their development, whereas the control cells bearing the OTI TCR developed exclusively as CD8 SP (Supplementary Fig. 9a, b). Taken together, these data show that although all tested clonotypes passed selection and developed as mature T cell cells in the retrogenic ATOC setup, only the PILT1 clonotypes developed a sizable CD44hi fraction with

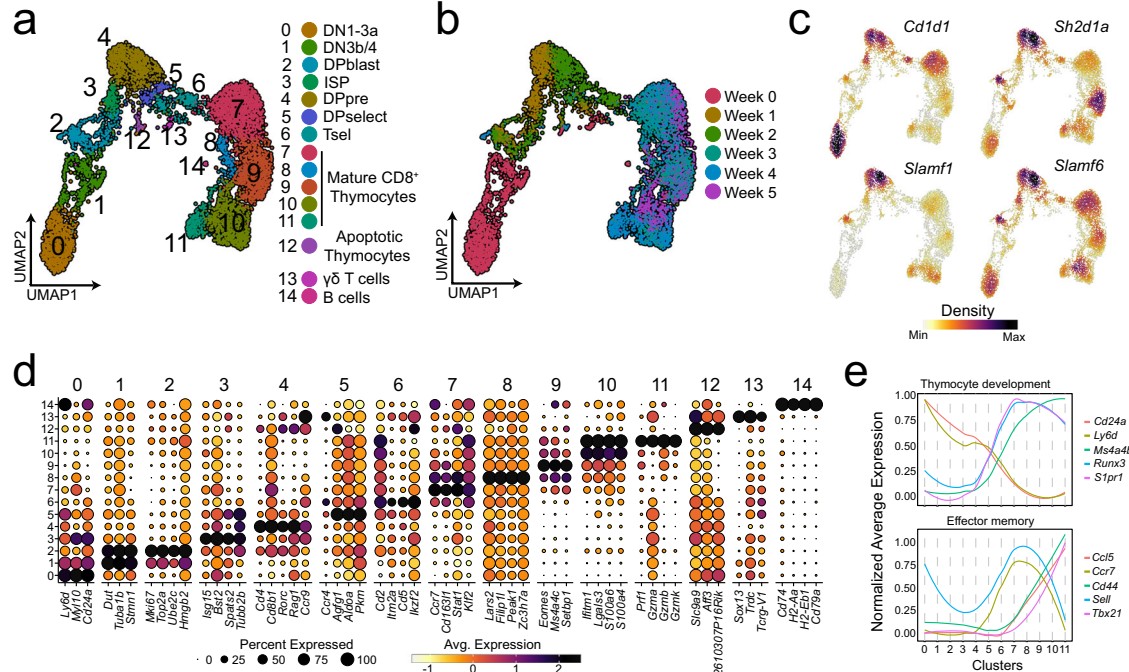

**Fig. 6 | Transcriptional landscape of ATOC-generated thymocytes.** UMAP of sorted DN thymocytes as shown in Supplementary Fig. 7b (week 0) and ATOC-generated αβ thymocytes from weeks 1:5 (identified as total seq γδ TCR negative cells) after QC n= 6884cells grouped by clusters (**a**) and time in weeks (**b**). DN1-3a is an abbreviation for double negative 1-3a stage thymocytes; DN3b-4 for double negative 3b-4 stage thymocytes; DPblast for double positive blast; ISP for immature single positive; DPpre for double positive pre-selection; DPselect for double positive entering positive selection and Tsel for thymocytes undergoing positive selection. **c** Density plots for selected gene markers associated with innate-like T cell selection and development for cells in Fig. 6a. **d** Dot plot showing the scaled log normalised average expression of the top 5 differentially expressed gene markers for each cluster in Fig. 6a. **e** Changes in normalised average expression of selected genes in clusters from Fig. 6a.

similar innate-like phenotype to their counterparts developing in vivo in the thymus[26].

## Discussion

In a recent study, we showed that PILT cells exhibit similar phenotypical characteristics to other members of the innate-like T cell family such as iNKT cells. In particular, PILT cells were found to differentiate in the same effector subsets as iNKT cells, however in contrast to iNKT cells, the initial screen for V-beta chain usage suggest a rather diverse TCR repertoire utilized by PILT cells[26]. In the present study, we aimed to provide a comprehensive view of the transcriptional and the TCR repertoire landscapes of PILT cells at a single cell level. Using scRNAseq approach we were able to build an in silico developmental trajectory, predict MHC-restriction, identify possible biases in the TCR repertoire and investigate the instructive role of PILT TCR specificity during PILT-cell selection and development.

The lack of unique surface markers expressed by PILT cells drove us to design a suitable gating strategy allowing enrichment for thymocytes with antigen-experienced phenotype, including PILT cells, which can be further processed for scRNAseq analyses. Following this approach, we analyzed thymi from MHC-T mice in which PILT cells are increased in frequencies due to conditional upregulation of MHC-I in the DP stage. In parallel, we used the thymi of CD1d$^{-/-}$ mice to get an overview of PILT cells when DP thymocytes have a base-level expression of MHC-I as in the WT scenario (Supplementary Fig. 10a, b)[64].

Here we show that by using our gating strategy, we were able to capture all three major effector subsets of PILT cells alongside PILT cell progenitors. In direct comparison, we show that PILT and iNKT cells have similar transcriptional profiles and follow a similar development pathway. Moreover, similar to what was recently reported for iNKT cells[14,16,18,20], our analysis revealed that thymic derived PILT cells exhibit further heterogeneity within the PILT1 and PILT2 cell subsets.

Interestingly, PILT cells had a higher PILT0 frequency in the thymus and a prominent CCR7$^+$ PILT2a fraction, which is suggestive of a higher rate of thymic generation of PILT cells in comparison to iNKT cells. However, we have previously reported that under normal steady-state conditions, PILT cells in the thymus are found in lower frequency than iNKT cells. One possibility that might explain the low PILT cell frequency is a high egress rate of PILT cells from the thymus into the periphery, which can be further validated by investigating recent thymic emigrants (RTEs) cells either by using intrathymic injection approach[65] to deliver labeling agents directly into the thymus or a RAG2p-GFP mouse models, both allowing tracking RTEs in periphery. Additionally, it is currently unknown what is the half live of PILT cells and what are their homing properties? Does PILT cell population or a sub-population thereof expand with age and do they possess a self-renewal potential? These are all characteristics that may help explain the low frequency of PILT cells observed thus far. Regardless, it appears that PILT cells are kept as an immunological backup in low frequencies and currently there is no approach to provoke their expansion in experimental disease settings in mice, thus providing functional insights.

We previously demonstrated that upregulation of MHC I at the DP stage of thymocyte development results in a significant increase in absolute cell-numbers of each PILT1/2/17-cell subsets in the thymus and periphery[26]. Notably, we also showed that the PILT-cell subsets ratio differs between WT and MHC-T mice with a marked decrease in PILT17-cell frequencies in the latter, a finding which we also confirm in the present study at a transcriptome level (Fig. 3c right panel). Noteworthy, we have also showed that PILT and iNKT cells compete for a cellular niche within the thymus[26]. Taken together, these results can be explained by two possible non-mutually exclusive scenarios. On the one hand, forced expression of MHC I molecules on DP thymocytes may favor development of PILT1 and PILT2-cell subsets and to a lesser

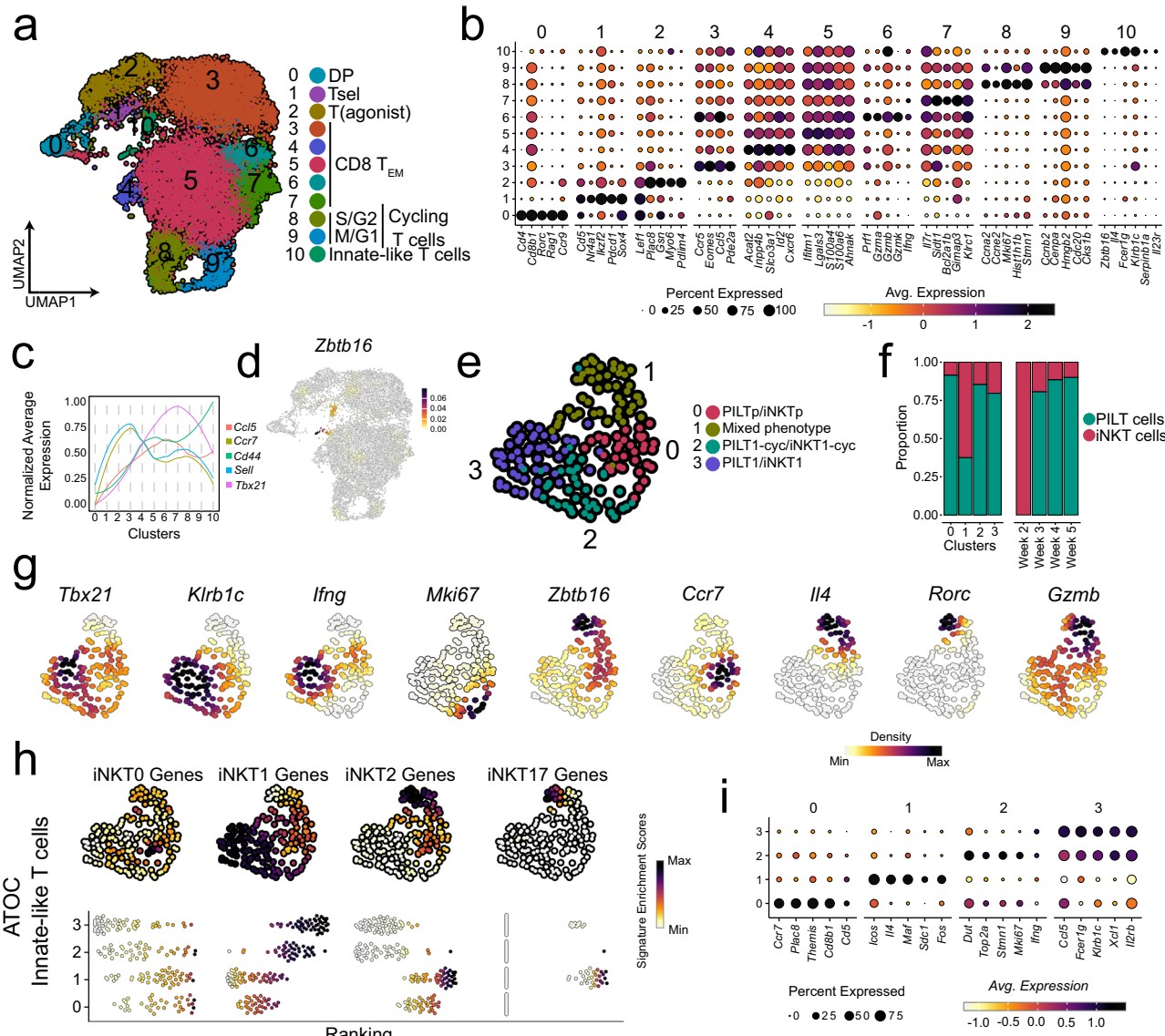

**Fig. 7 | Transcriptional landscape of ATOC generated innate-like thymocytes.**
**a** UMAP of sorted PILT cell-enriched population (as shown in Supplementary
Fig. 1a) from ATOCs at weeks 1:5 $n = 14,019$ cells. DP is an abbreviation for double
positive thymocytes; T(agonist) for agonist-signaled thymocytes; Tsel is for thy-
mocytes undergoing positive selection; CD8 $T_{EM}$ for CD8 effector memory T cells;
S/G2 for cycling T cells in S/G2-phase and M/G1 for cycling T cells in M/G1-phase.
**b** Dot plot showing the scaled log normalised average expression of selected genes
for cells in Fig. 7a. **c** Changes in normalised average expression of selected genes in
clusters from Fig. 7a. **d** Density plot of *Zbtb16* in Fig. 7a. **e** UMAP of ATOC-generated
innate-like thymocytes (cells from cluster 10 and cells with *Zbtb16* expression >0)

$n = 173$ cells. PILTp is an abbreviation for PILT-cell progenitors; iNKTp for invariant
natural killer T-cell progenitors; PILT1-cyc for cycling PILT1 cells and iNKT1-cyc for
cycling iNKT1 cells. **f** A filled bar plot showing PILT and iNKT cells distribution in
clusters (left) and time by weeks (right) from Fig. 7e. **g** Density plots of selected
genes in Fig. 7e. **h** Feature (top) and bee swarm (bottom) plots for KNN smoothed
signature enrichment score for iNKT subsets gene markers in Supplementary
Table 1 for PILT cells. **i** Dot plot showing the scaled log normalised average
expression of selected genes for cells in Fig. 7e. Source data are provided as a
Source Data file.

extent PILT17 cells. On the other hand, the reduced PILT17 frequency in
the MHC-PILT cell pool might be a result of a competition for a limited
cellular niche supporting PILT17-cell development in the thymus.

Scrutinizing the PILT-cell TCR repertoire, we show that PILT cells
have a polyclonal TCR repertoire exhibiting a comparable diversity to
conventional T cells in both thymus and blood. However, a further
analysis of CDR1/2 TCR regions found in PILT, CD8 and CD4 T cell
repertoire showed a higher similarity between PILT and CD8 T cells in
comparison to PILT and CD4 T cells. Moreover, our data revealed a
similar pattern of Vgene-α and Vgene-β usage by PILT and CD8 T cells
(Supplementary Fig. 4a, b). Taken together, these data suggest that
PILT cells are MHCI restricted since CDR1/2 regions govern the TCR
interaction with selected antigen presenting molecules. Notably, MHC-

PILT and WT-PILT share the top TCRα:TCRβ (TRAV9N-3:TRBV16) chain
pair however, said pair is present approximately four times more fre-
quently in the WT-PILT TCR repertoire than of that of MHC-PILT cells.
Interestingly, TRAV9N-3 have been previously described as chain used
by T cells with innate-like phenotype[66]. Whether there is an overlap
between PILT cells and other innate-like T cells is still an open question.
Therefore, these data suggest that PILT cells might harbor a small
innate-like T cells population restricted to non-classical MHC-I
molecule.

The overlapping transcriptional signature between PILT and
iNKT cells suggests similar functions shared between these innate-like
T cell populations. Akin to iNKT cells, PILT cells can produce the
cytokines interferon-γ (IFNγ), interleukin (IL)−4, and IL-17A as response

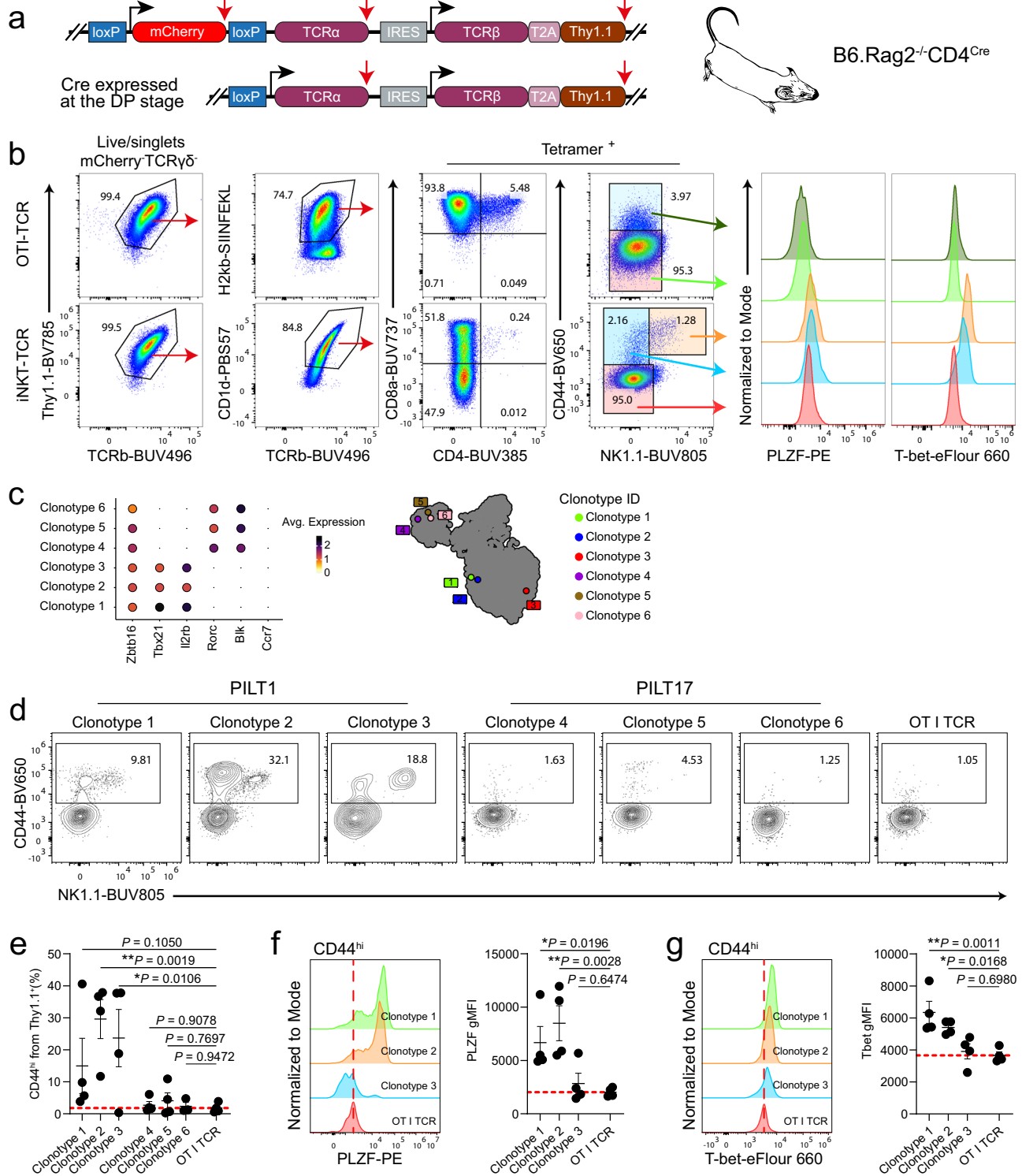

**Fig. 8 | T-cell receptor retrogenic T cells develop in the ATOC system.**
**a** Schematic drawing depicting the retroviral expression vector allowing "on-time" expression of TCRα in B6.Rag2$^{-/-}$CD4$^{Cre}$ settings. Black arrows indicate translation initiation sites and red arrows indicate stop codons. **b** Flow cytometry evaluation of T cell development in the T-cell receptor retrogenic ATOC system at week 4. Shown are representative plots of ATOCs seeded with the OTI TCR clonotype (Top) and a selected iNKT1 TCR clonotype (bottom). ATOCs were generated as depicted in Supplementary Fig. 6b. **c** Dot plot showing the log normalised average expression of selected genes by each of the PILT selected clonotypes (right) and its UMAP coordinates in Fig. 2a (left). **d** Representative flow cytometry plots showing CD44 and

NK1.1 expression on T cells generated in the T-cell receptor retrogenic ATOC system. ATOCs were generated as depicted in Supplementary Fig. 6c. Cells are gated on live/ singlets/Thy1.1$^+$. CD44$^{hi}$ cell frequency are shown in (**e**). Flow cytometry evaluation of PLZF expression in (**f**) and T-bet expression in (**g**) on CD44$^{hi}$ T-cell receptor retrogenic PILT1 cells. Each point represents one ATOC in (**e**–**g**): $n = 5$ ATOCs per group in (**b**) and $n = 4$ ATOCs per group in (**d**–**g**). Data are representative of three in (**b**) and two in (**d**–**g**) biologically independent experiments. Statistical significance was calculated using one-way ANOVA followed by Fisher's LSD multiple comparisons test in (**e**–**g**); not significant ($P \geq 0.05$), *$P < 0.05$ and **$P < 0.01$. Data are presented as mean values ± SEM. Source data are provided as a Source Data file.

to in vitro stimulation[26]. Moreover, PILT1 and iNKT1-cell subsets share high expression levels of perforin (*Prf1*), granzyme a (*Gzma*) and granzyme b (*Gzmb*), Fas ligand (*Fasl*) and TRAIL (*Tnfsf10*) inferring similar killing capacity of both subsets (Supplementary Fig. 2b). Further, we previously reported that akin to iNKT2 cells, IL-4 production by PILT2 cells promotes CD8+ Memory-like T cell development in the thymus. Lastly, You et al. recently described a novel function of innate-like T cells by showing that innate-like T cells in the thymus can mediate CD8+ T cell tolerance induction to T cell-derived inflammation-associated self-antigens such as IL-4, IL-17A and granzyme b[67]. In that study, the authors showed that this function was supported by redundancy in cell types expressing these molecules including a noticeable fraction of non-iNKT αβ T cells resembling PILT cells. Overall, these data infers a level of redundancy in the functional role of iNKT and PILT cells, yet in contrast to iNKT cells, PILT cells utilize a polyclonal TCR repertoire with an unknown antigen specificity.

Sterile production of type I and type III interferons (IFNs) in the thymus was recently shown to play a role in shaping Treg TCR repertoire diversity[68]. Interestingly, some MHC I expressing DP thymocytes co-express IFN-stimulated genes (ISGs) such as *Isg15*, *Isg20* and *Ifi203*, indicating an IFN signaling signature (Supplementary Fig. 10c). Therefore, it is possible that IFNs production in the thymus might impact the diversity of the PILT-cell TCR repertoire by facilitating selection of PILT cell clonotypes restricted to self-antigens such as ISGs. In the hypothesized scenario, PILT cells would serve as responders in the initial stages of viral infections.

Different setups of in vitro ATOC systems are widely used to study stages of thymopoiesis in in vitro settings. Here, we validate previous findings that ATOCs support the development of αβ thymocytes and further show at transcriptome level that developing DN thymocytes in the ATOC system go through the same developmental stages as their counterparts in the thymus. Notably, akin to DP thymocytes developing in the thymus, ATOC-derived thymocytes at the DP stage were found to express *Cd1d1* and the co-stimulatory *Sh2d1*, *Slamf1* and *Slamf6* molecules rendering them capable of selecting innate-like T cells. Indeed, a small fraction of PLZF+ innate-like αβ T cells emerged as early as week 1 from the start of the organoid cultures. However, these cells upregulated CD44 and NK1.1 expression only at later time points (at week 3) excluding the possibility of contamination with mature innate-like T cells during the cell sort of the seeding DN thymocyte population. Surprisingly, a subsequent scRNAseq analysis of ATOC generated innate-like T cells revealed that, although this system supports the development and differentiation of PILT1 and iNKT1 cells, we could not identify cell populations with distinct PILT2 or PILT17 phenotype but only a rather small cell-fraction exhibiting a mixed phenotype. Therefore, these data infers that there are missing instruction signals needed for PILT2 and PILT17 development. Whether such signals are in the form of cytokines, cognate antigens or interacting partners of myeloid origin[69] still remains to be answered yet, the utilized ATOC system in this study can serve as versatile tool to address such a question in future studies.

In a next step, by utilizing a retrogenic ATOC system allowing an "on-time" expression of a defined TCR clonotype at the DP stage, we addressed the question whether TCR specificity plays a role in PILT cell linage commitment and differentiation. Firstly, we showed that DN thymocytes bearing a selected innate-like TCR clonotype (iNKT1-cell derived) were able to give rise to a small fraction of PLZF+ T cells with a mature iNKT1 phenotype in the course of 4 weeks developing in the ATOC system. In contrast, ATOCs seeded with DN thymocytes bearing a conventional TCR clonotype (such as OTI TCR) did not give rise to PLZF+ T cell population but rather developed as conventional T cells. In this system, DP thymocytes serve as selection partners while expressing both antigen-presenting molecules (CD1d and H2-Kb for iNKT TCR and OTI TCR respectively), however only CD1d could be loaded with a presumably endogenous cognate antigens providing a strong enough TCR signaling for an agonist selection to occur. Therefore, while H2-Kb expression by the DP thymocytes was sufficient for OTI T cell selection and development the absence of the cognate antigen (SIINFEKL) in the system did not facilitate a strong TCR singling required for agonist selection of OTI T cells with innate-like phenotype.

Lastly, by tracking the development of a selected set of three PILT1 and three PILT17 clonotypes in the retrogenic ATOC system, we were able to show that only organoids seeded with thymocytes bearing PILT1 TCR clonotypes developed a substantial fraction of cells with a PILT1 innate-like T cell phenotype, whereas organoids seeded with thymocytes bearing PILT17 TCR clonotypes developed as mature T cells but did not acquire an innate-like phenotype. Expectedly, these results corroborate the notion that the current ATOC system cannot support development of PILT17 cells, however the fact that the PILT1 TCR clonotypes favored development of prominent innate-like T cell fraction with PILT1 phenotype is suggestive that TCR specificity play an instructive role in PILT cell development and subset differentiation.

In the present study, we pinpoint the transcriptional and developmental similarities between PILT cells and other members of the innate-like T cell lineage. Moreover, we provide valuable insights into the unique characteristics of PILT cell TCR repertoire predicting their MHC restriction and level of diversity. Interestingly, our study identified TCRs that sponsor the commitment of developing thymocytes to the PILT cells lineage raising the question, what antigens are recognized by these TCRs? Essentially, identification of putative antigens will be instructive in understanding the role of these cells in immune-relevant challenges in the long run also in humans.

## Methods

### Mice
C57BL/6 N (B6) mice were obtained from Charles River. B6.Cg-Tg(Cd4-cre)1Cwi/BfluJ (CD4^Cre strain: #022071), B6.129S6-Del(3Cd1d2-Cd1d1)1Sbp/J (CD1d^−/− strain: #008881), B6.129P2-B2m^tm1Unc/J (B2m^−/− strain: #002070) and B6.Cg-*Rag2*^tm1.1Cgn/J (Rag2^−/− strain: #008449) mice were purchased from the Jackson Laboratories. The B6.-Gt(ROSA)26Sor^tm1(CAG-Nlrc5)Khog (Nlrc5-stop^flox MGI: #7286188) mice were kindly provided by Kristin Hogquist (University of Minnesota, Minneapolis). All mouse strains used in this study are on B6 genetic background. All animals used in this study were 8–12 weeks old at the time of analysis. All mice were maintained under specific pathogen-free conditions in the animal facility of Hannover Medical School under a 12 h light/dark cycle with ad libitum access to food and water, ambient temperature between 22 and 24 °C and humidity between 45 and 65%. All mice used in this study were euthanized via CO_2 inhalation method. All experimental procedures were conducted in accordance with the local animal welfare regulations reviewed by the institutional review board and the Niedersächsisches Landesamt für Verbraucherschutz und Lebensmittelsicherheit (LAVES) under the permission 2022/306.

### Flow cytometry
Single-cell suspensions were prepared on ice in FACS buffer (PBS/3% fetal bovine serum). All surface stainings were performed In FACS buffer on ice for 30 min. Intracellular staining for T-bet and PLZF was done using eBioscience FOXP3 / Transcription Factor Staining Buffer Set (Invitrogen, Cat: 00-5523-00), according to the manufacturer recommendations. All antibodies used in this study are listed in Supplementary Table 2. CD1d tetramers loaded with PBS57 (analog of a-galactosylceramide) and MR1 tetramers[70] loaded with 5-OP-RU were provided by the NIH Tetramer Core Facility. Flow cytometric analysis was performed on Cytek Aurora (Cytek Biosciences) using SpectroFlo software (version 3.3.0, Cytek Biosciences) and data analysis was done using FlowJo software (version 10.8.1, BD Biosciences).

## Cell culture

MS-5 cell-line was purchased from (Leibniz Institute DSMZ-German Collection of Microorganisms and Cell Cultures GmbH). Lenti-X cells were purchased from (Takara Bio, Cat: 632180). Both cell lines were maintained in a complete DMEM medium DMEM supplemented with 10% Fetal Bovine Serum (FBS), 1%Penicillin-Streptomycin (Gibco, Cat: 15140122, 1%HEPES (Merck, Cat: H0887-100ML) and, 1%GlutaMax (Gibco, Cat: 35050061)).

## Cloning

Full-length codon optimized coding sequences (CDS) of mouse mDLL1 (NM_007865.3), mDLL4 (AF253469.1), mH2-Kb (U47328.1) and mH2-Db (U47325.1) along with CDS of selected TCRα and TCRβ chains from PILT and iNKT clonotypes were ordered separately as gBlocks (IDT). NEBuilder HiFi DNA Assembly kit (NEB, Cat: E2621X) was used for cloning the gene blocks into pMig retroviral expression plasmid kindly provided by Kristin Hogquist (University of Minnesota, Minneapolis). Assembly reaction products were then used to transform Mix & Go! Competent Cells - DH5 Alpha (Zymo research, Cat: T3009). Single colonies were then picked and further verified by DNA sequencing. All CDS used in this study are listed in Supplementary Table 3.

## Viral vector production

All retroviruses were generated by transfection of Lenti-X cells (Takara Bio, Cat: 632180) with pMig expression plasmid containing the gene inserts along with pCL-Eco packaging plasmid. Transfection was carried out using Calcium Phosphate Transfection Kit from (Invitrogen, Cat: K278001) according to the manufacturer recommendations. The following day the medium was replaced with fresh complete DMEM medium supplemented with ViralBoost (Alstem, Cat: VB100). The day after the supernatant containing the retrovirus was collected and centrifuged at 1000 x $g$ for 10 min and then filtered through a 0.45 μm filter (Roth, Cat: KH55.1). The filtrate was then used for transduction.

## Generation of modified MS-5 cell lines

MS-5 cells (Leibniz Institute DSMZ-German Collection of Microorganisms and Cell Cultures, Cat: ACC 441) were plated into 6 well plates with a concentration of $1 \times 10^5$ cells per well. On the following day the medium was replaced with virus-containing supernatant supplemented with 4 μg/ml Polybrene (Merck, Cat: TR-1003-G) and further incubated at 37 °C for 8 h. Following incubation, virus-containing supernatant was exchanged with a fresh complete DMEM medium. After three passages, transduced cells were evaluated by flow cytometry for transduction efficiency and expression level of the target proteins. The highest 10% target gene-expressing cells were sorted by FACS.

## Artificial thymic organoids

Artificial thymic organoids were generated according to a modified protocol based on previous publications[62]. In brief, single-cell suspensions of thymocytes were prepared at room temperature and then incubated with antibody mix (Supplementary Table 2) in a staining buffer for 20 min. MojoSort Streptavidin Nanobeads (BioLegend, Cat: 480016) were then used as per manufacturer recommendations to enrich DN thymocytes by depleting antibody mix positive cells. Enriched DN thymocytes were then incubated with the sorting antibody mix (Supplementary Table 2) and sorted on FACSAria Fusion (Becton Dickinson). For each organoid, 1 mL of organoid medium DMEM-F12 supplemented with 2% B27 supplement (Gibco, Cat: 17504-044), 30 μM L-ascorbic acid 2-phosphate sesquimagnesium salt hydrate (Merck, Cat: A8960-5G), 1% Penicillin-Streptomycin, 1% Glutamax, 5 ng/ml rmFLT3L (PeproTech, Cat:250-31 L), 5 ng/ml rmIL-7 (PeproTech, Cat:217-17), 10 ng/ml rmSCF (PeproTech, Cat:250-03) (only for first week) and, 0.05 mM beta-mercaptoethanol (Merck, Cat:M7522) was transferred into a well from 6-well plate. A 0.4 μm Millicell transwell

(Merck, Cat:PICM0RG50) insert was then laid into each of the wells. Per organoid, $5 \times 10^4$ DN thymocytes were mixed with $15 \times 10^4$ MS-5 cells, centrifuged at 300 x $g$ for 5 min and resuspended in 5 μl of organoid medium. The cell mix was then transferred on top of the Millicell transwell insert. The culture medium was replaced every three to four days. Organoids were resuspended into organoid collection buffer (PBS/0.5% bovine serum album/2 mM EDTA) which were later passed through a 40 μm strainer.

For retrogenic ATOC organoids, DN thymocytes from B6.CD4$^{Cre}$Rag2$^{-/-}$ mice were transduced with different retroviral clonotype constructs by spinfection at 1000 x $g$. for 90 min at 37 °C. Transduced cells were then mixed with DN thymocytes from MHC-T mice (1:5) or used alone to seed the organoids as indicated in Supplementary Fig. 6b, c.

## Cell isolation for single cell RNA sequencing experiments

Single-cell suspensions of thymocytes were prepared on ice in staining buffer and then stain with a rat anti-mouse CD8α antibody (clone: CD8alpha; in-house produced) for 30 minutes on ice. Dynabeads Sheep Anti-Rat IgG (Invitrogen, Cat: 11035) were used to deplete CD8$^{pos}$ thymocytes as per the manufacturer recommendations. Depleted fraction was used in experiment 1 and GEM 2 in experiment 2. No depletion was done in the rest of experiments. Next, cell-suspensions were processed for surface staining with an appropriate antibody mix (Supplementary Table 2) and sorted on FACSAria Fusion (Becton Dickinson). Cell fractions were collected as follows: iNKT (Live/CD19$^-$/TCRβ$^+$/CD1d-PBS57$^+$) and PILT cells enriched population (Live/CD19$^-$/TCRβ$^+$/CD1d-PBS57$^-$/TCRβ + /CD44$^+$ and/or PD-1$^+$) for experiments 1, 2, 3 and 5 or total live cells for experiment 4 (Supplementary Fig. 1b, c and Supplementary Table 4). All cell sorts were done on FACSAria III Fusion (Becton Dickinson) and the purity of sorted cells was routinely >95%.

## 10x single-cell gel bead-in emulsions (GEM) generation, library preparation, and sequencing

**Experiment 1**. Single-cell suspensions of thymocytes were prepared from 3 WT mice and 3 MHC-T mice (10 weeks old). All samples were subjected to CD8α depletion, stained with appropriate antibody mix followed by FACS sorting. Sorted cells were mixed in a ratio of 1:4 (iNKT:PILT enriched population) and 20,000 cells were used for GEM1 generation.

**Experiment 2**. Single-cell suspensions of thymocytes were prepared from 3 WT mice and 3 MHC-T mice (10 weeks old) and then split into two equal parts. Part 1 was depleted for CD8α (as in Experiment 1) and Part 2 was not. Both parts were stained with appropriate antibody mix followed by FACS sorting. Sorted cell fractions were then mixed in a ratio of 1:4 (iNKT:PILT-enriched population) and 40,000 cells were used for GEM2 (CD8α depleted) and GEM3 (non-deplete) generation.

**Experiment 3**. Single-cell suspensions of thymocytes were prepared from 3 CD1d$^{-/-}$ mice (10 weeks old), stained with appropriate antibody mix followed by FACS sorting. Sorted cells were then mixed and 40,000 cells were used for GEM4 generation.

**Experiment 4**. For samples week1:5, we collected and pooled three organoids per time point in organoid collection buffer. The sample was then split into two equal parts. Part 1 was stained with an appropriate antibody mix and subjected to FACS sorting for iNKT cells (Live/CD19$^-$/TCRβ$^+$/CD1d-PBS57$^+$). Part 2 was stained with an appropriate antibody mix and subjected to FACS sorting for PILT-enriched population (Live/CD19$^-$/TCRβ$^+$/CD1d-PBS57$^-$/TCRβ + /CD44$^+$ and/or PD-1$^+$). All sorted iNKT cells were then pooled with 40,000 cells from the PILT-enriched population and used for GEM6 generation.

For experiments 1, 2, and 3 the GEMs were generated by Chromium controller (10x Genomics) using Chromium Single Cell V(D)J

Reagent Kits (v1.1). For experiments 4 and 5 the GEMs were generated by Chromium Xi (10x Genomics) using Chromium Single Cell 5' Reagent Kits (v2−Dual Index). The library preparation was carried out as per manufacturer recommendations for each of the kits. Generated sequencing libraries were either sequenced by Illumina's NextSeq 550, NovaSeq 6000 or NovaSeqX with an average read per cell of 41452 for gene expression libraries and an average of 16794 and 8028 reads per cell for VDJ and surface libraries respectively. Schematic representation of the layout of all single-cell RNA sequencing experiments is depicted in (Supplementary Fig. 1b,c).

**Experiment 5**. For week 0 sample, FACS sorted DN1-3 thymocytes from 3 WT mice (10 weeks old) were used as shown in (Supplementary Fig. 7b). Samples for weeks 1:5 were collected from the corresponding organoids in an organoid collection buffer and then stained using Zombie Green Fixable Viability Kit. FACS sorted live thymocytes from samples from weeks 0:5 were then mixed in equal numbers and 40,000-pooled cells were then used for GEM5 generation.

### Bioinformatic analysis
Generated FASTQ files were processed with Cell Ranger (v7.0.1) Multi pipeline using the default parameters (intronic reads included). Seurat[71] (v4.3.0.1) was used to import the Gene/Hashtag count matrices for each of the experiments. VDJ information was later added to Seurat objects as metadata using djvdj[72] (v0.1.0). To clean the dataset we removed the ambient RNA counts using SoupX[73] (v1.6.2) and cells with high mitochondrial gene percentage. Cells with MAIT TCR or no TCR information were also removed from the analysis. Hashtag demultiplexing and doublet removal were done using HTODemux[74] function from Seurat. *Trav*, *Trbv* and *H2* genes were removed from the datasets so as not to influence downstream analysis. Data was normalised and scaled following Seurat default recommendation. Latent representations computed by scVI[75] (v1.0.0) based on the top 2000 variable genes were used to build the integrated UMAP. Differentially expressed gene between clusters were identified Surat's findallmarkers function. Density plots were generated using Nebulosa[76] (v1.0.1). Gene signature scores were calculated using UCell[77] (v2.4.0). Pseudo-time analysis was performed using Monocle3[78] (v1.3.1). For gene set enrichment analysis, deferentially expressed genes (DEGs) between clusters were identified using wilcoxauc function from presto[79]. DEGs for PILT0 and iNKT0 clusters were ranked based on logFC values. A custom human gene set with "CD8 OR CD4 OR DP OR Thymocytes OR TCELL OR T_CELL" keywords were downloaded from "The Molecular Signatures Database". Babelgene library was used to transform the human custom gene set to its mouse ortholog. The GSEA function from clusterProfiler[80] (v4.12.0) was used for the gene set enrichment analysis. Before integrating PILT cells with the cell atlas of human thymic development we changed the gene name in PILT cells dataset to its human ortholog using biomaRt[81] (v2.62.1). Common genes between the two datasets were used for the downstream analysis.

### Visualization and figures generation
Single-cell data visualisation was done using ggplot2 (v3.4.3), SCpubr[82] (v2.0.1), and djvdj[72]. FACS plots were generated in FlowJo (version 10.8.1, BD Biosciences). Figures were made using Inkscape (v1.3.2) and Adobe Illustrator (v15.1.0 and v29.3).

### Statistics and reproducibility
A seed was set to "44" for R, py and clusterProfiler and was set to "20" for scVI to ensure reproducibility. Statistical analyses were performed using R (v4.3.1) and/or GraphPad Prism (v8). Mast method was used in findallmarkers function to identify DEGs using cellular detection rate as a covariate with the following parameter min.pct = 0.25, logfc.threshold = 0.25, only.pos = TRUE. GESA

function from cluster profiler was used with eps = 1e−300, min-GSSize = 20, $p$ valueCutoff = 0.05, nPermSimple = 10,000, seed = 44. Spearman's rank correlation coefficient was used to assess the correlation between the different clusters/celltypes. Simpson's Diversity Index was used to assess the TCR gene usage and CDR3 region diversity. Horn-Morisita index was used to assess the CDR1/2 similarity between different cellstypes. FACS data are summarized as means with error bars showing the SEM. For FACS data, the statistical significance was assessed based on a one-way ANOVA followed by Fisher's LSD multiple comparisons test as indicated in the legends to the figures. $p$ values of <0.05 (*), <0.01 (**), <0.001 (***) or <0.001 (****) indicated significant differences between groups.

### Reporting summary

Further information on research design is available in the Nature Portfolio Reporting Summary linked to this article.

## Data availability
The data that support the findings of this study are provided either in Supplementary Figs. or in the Source data file. The raw data and cell-ranger outputs generated in this study have been deposited in the GEO repository under accession code GSE279513. All other data are available in the article and its Supplementary files or from the corresponding author upon request. Source data are provided with this paper.

## Code availability
No custom code was generated in this study and the default parameters were used for all arguments unless otherwise mentioned.

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

## Acknowledgements

We thank all present and past members of the Georgiev lab and members of the Institute of Immunology at the Hannover Medical School for the inspiring discussions and assistance. We also thank Dr. Günter Bernhardt and Dr. Kristin Hogquist for critical reading of the manuscript. We wish to thank the team from the sorter unit and the team from the Research Core Unit Transcriptomics (RCUT) at Hannover Medical School for invaluable support. We thank Svetlana Piter for providing excellent animal care. We acknowledge the NIH Tetramer Core Facility for provision of mCD1d and mMR1 tetramers[70]. This work was supported by Deutsche Forschungsgemeinschaft (DFG, German Research Foundation) grant number GE3062/2-1 to H.G and Excellence Strategy EXC 2155 "RESIST" project ID 390874280 to R.F. A.H. and N.H are supported by the Hannover Biomedical Research School (HBRS) and the Center for Infection Biology (ZIB).

## Author contributions

H.G., R.F. and A.H. conceptualized the work. H.G and A.H. designed experiments; H.G., A.H. and N.H. performed experiments; H.G., A.H. and N.H. analyzed experiments and interpreted the findings. A.S., S.W. and I.R. helped with genotyping and scRNAseq libraries preparation. A.H. wrote the manuscript. H.G. and R.F. edited the manuscript. H.G. directed the research and is the guarantor of its integrity. All authors contributed to manuscript revision and approved the submitted version.

## Funding

## Competing interests

The authors declare no competing interests.
