## [Transparent Peer Review file · Nature Communications]

Developmental features and unique characteristics of peptide-specific PLZF⁺ innate-like T cells in mice

Corresponding Author: Dr Hristo Georgiev

Version 0:

Reviewer comments:

Reviewer #1

(Remarks to the Author)

This study presented by Hassan et al. explores the development of peptide-specific PLZF⁺ innate-like T (PILT) cells using single-cell transcriptomic analysis. The researchers found that PILT cells share transcriptional profiles with invariant Natural Killer T (iNKT) cells but have a polyclonal TCR repertoire, suggesting MHC I restriction and broader antigen specificity. They also showed that artificial thymic organoid cultures (AOTC) support PILT cell development and that TCR specificity influences their lineage commitment and differentiation.

This paper presents a comprehensive collection of data and analyses that provide valuable insights into the transcriptional profile of a distinct subset of innate-like T cells. These findings are well-positioned to enhance the field's understanding of innate-like T cells and how their subsets compare and contrast. However, there remain some technical questions that require further clarification, and a recommendation that this body of work may belong in a setting that is more appropriate for the amount and type of data that is provided, such as a Resource paper in an immunology journal.

Critiques:

- It is unclear whether cells were pooled for the scRNAseq experiments described in Extended Fig. 1b, or how this process was carried out. Additional details are needed in the narrative, or a clearer diagram should be provided to explain how this experimental design resulted in the UMAPs shown in Fig. 1.
- Authors do not provide any absolute numbers that describe the cell populations of interest, only percentages, which does not create a complete picture of what might be happening with other cell types.
 - o For example, in Figure 5c and d, authors are making a claim that in this AOTC model there is an emergence of PILT cells and iNKT cells, but do not provide the absolute cell numbers over time. There could be changes in other cell populations that influence the percentage of PILT cells without the PILT cell population actually increasing.
- There is no discussion of what the function of these cells might be and why they might be important to study. How would these cells respond to activation in vitro? What cytokines are they capable of producing? Is there any evidence that suggests that their function may differ from other innate-like T cells function given that they are transcriptionally similar?
- As mentioned in the text of this paper, this group has already published about the identification of the PILT cell subset and this paper seems to be building off of the previous publication, characterizing its transcriptional profile, without actually concluding anything novel about these cells. Because of this, this work seems to lend itself better as a resource rather than as a standalone research article.

Reviewer #2

(Remarks to the Author)

In this manuscript, Georgiev et al. examine PLZF⁺ MHC class I-restricted T cells (PILTs) in the thymus.

PLZF expression in the thymus occurs in NKT cells and other innate-like T cells such as MAIT cells. A key difference in thymic selection between conventional T cells and NKT cells is that conventional T cells are selected by thymic epithelial cells, while NKT cells get selected by adjacent double positive (DP) thymocytes. This occurs, because DP thymocytes in WT mice express CD1d, but not conventional MHC class I or II. In a previous study, the authors generated mice that forced expression of MHC class I on DPs in CD1d deficient mice and observed that this led to the generation of MHC class I-restricted PLZF⁺ T cells (PILTs). In this manuscript, the authors further examine these PILTs using scRNAseq (on ex vivo

isolated cells) and thymic organoid cultures.

Overall, the experiments are well designed and well described to build a very coherent and thorough body of work. My main question is in regards to the conclusions and the biological relevance of these PILTs. The authors seem to suggest that PILTs are a naturally occurring T cell lineage (the last sentence in the discussion even suggests that these cells occur in humans).

However, generation of these cells requires the experimentally forced expression of MHC class I on DPs (T MHC I mice). I did not see any compelling evidence in this manuscript or the initial report indicating that PILTs develop in WT mice. Since WT DPs do not express MHC class I, how could PILTs even develop in WT animals?

The intriguing findings to me were related to the PILT – NKT comparisons made by the authors, which do provide very unique and novel insight to the potential role of CD1d/antigen vs. DP-mediated selection in the functional development of NKT cells.

In its current state the manuscript is rather confusing, because the key conclusion is not really clear. If the authors truly want to argue that PILTs occur in a WT environment (and the T MHC I mice are merely a tool to increase PILT numbers to enable scRNAseq, etc. experiments) then additional experiments are required to support this notion. If the intended main focus is on using PILTs to improve our understanding of NKT cell development, then editing for clarification is sufficient. Finally, I do want to emphasize again that this is a very compelling, creative and intriguing set of experiments that provides unique insight into thymic selection and cell fate decision processes.

Reviewer #3

(Remarks to the Author)

In this study, Hassan et. al., nicely describes the transcriptional profiles and potential developmental features of antigen-specific PLZF+ innate-like T cells compared with iNKT cells, utilizing multi-modal single cell sequencing (RNA + TCR) as well as an in vitro thymic organoid culture system. This paper is well-written and provides unique insight into this newly described T cell type, its potential broad functionality and its development. However, the study has some gaps that should be addressed to clarify the conclusions and strengthen the overall impact of the findings.

Main comments:

1. Based on the lack of unique surface markers to identify PILT cells, authors used a simple strategy to enrich more PILT cells. Using *Zbtb16* (PLZF gene) expression as a marker for this population in their scRNAseq clustering, they found a wide range in *Zbtb16* expression and, what looks like an overall lower gene expression in PILT-1 cells, across experiments. At a protein level, there also seems to be a range of expression in the PILT population. It would be easy to confirm that PLZF-low PILTs are transcriptionally bias towards PILT-1 type subsets and would overall help the clarity of the study; and that these populations are really the expected PILTs.
2. It is somewhat surprising that PILT and iNKT cells completely overlap in Fig 3. How were these umaps harmonized? Are there unique RNA features that distinguish this population from iNKT beyond TCR expression?
3. The distribution of the PILT cell states (1, 2, 17 etc) is different between WT and T-MHC mice. (Fig 3 c, 2nd plot) This is a very interesting finding and may provide some insight into alterations in PILT development. Authors should discuss this in more detail.
4. Some of the most interesting findings were those in the transgenic ATOC system, however authors have limited discussions on these findings. Of particular interest are: 1) There is marked loss of PILT cells in the ATOC system at 5 weeks. (Fig 5) Why do these cells develop and disappear? Is it technical or biological? 2) Only two out of the three PILT1 TCR clones exhibit PLZF expression. (Fig 8) Why might this be?
5. On a broader level, age plays an important role in the innate-like properties of T cells. (Reynaldi, PNAS, 2019; Smith, Cell, 2018) Does the mouse age used in these studies, or the thymic stromal cell culture "age" play a role in the frequency or "state" (1, 2, 17 etc) of PILT? What about other basic features such as sex or mouse strain?
6. How do these new innate-like T cells in mouse related to the more innate-like populations recently identified in human blood? (Thomson, Nat Imm, 2023; Terekhova, Immunity, 2023) Human thymic datasets could be used to compare overall transcriptional similarities between mouse and human subsets (Park, Science, 2020).

Minor comments:

1. The distribution umap plots in Figure 3 are hard to understand as presented, especially 3b, 3g and 3i.
2. Experiments were ran across a number of different sequencing runs, which can impact data quality (# of cells per samples, read counts per cells etc). Are there any batch effects in these data? It would be good to provide sample QC metrics for these datasets.
3. The coloration used for gene expression is not consistent throughout the manuscript. It would be helpful to keep this

consistent (e.g. black is always low, white is always high)

4. Fig 5c, 8e, 8g: There are no basic stats on these graphs. P-values should be included.

5. Fig 8 c – cannot tell the colors of the clone in the umap.

Version 1:

Reviewer comments:

Reviewer #1

(Remarks to the Author)

The authors have provided new data and revised the manuscript. The revised version has adequately addressed the concerns raised about the original version.

Reviewer #2

(Remarks to the Author)

I stated in my initial review that “If the authors truly want to argue that PILTs occur in a WT environment (and the T MHC I mice are merely a tool to increase PILT numbers to enable scRNAseq, etc. experiments) then additional experiments are required to support this notion.”

The authors make it clear in the response that they indeed argue PILTs occur in a WT environment. I am somewhat surprised by this since in the initial report Georgiev and Hogquist concluded (last sentence of the abstract): “We demonstrate that PIL and NKT cells compete for a narrow niche, suggesting that the absence of peptide-MHC on DP thymocytes facilitates selection of non-peptide specific lymphocytes.”

Now, the authors are stating that peptide-MHC are in fact NOT absent on DP thymocytes. This is an important distinction from the initial report and a very different conclusion. There are 2 bare minimum experiments that they need to do to support this new conclusion:

1. Use flow cytometry to show which DPs express MHC class I (in context of CD1d expression) with proper staining controls (FMO). The expression pattern & levels in WT vs T MHC I mice should be compared and discussed as they are pertinent for the observed skewing in PILT 1 vs 17.
2. In Figure 1d of their response letter, the authors still see a few PILT cells in b2m^{-/-} mice. I assume the authors would argue that those events are “background”. I would agree with this in principle (PLZF antibody that didn't get washed out after the intracellular stain), but since the population is so rare, it is essential to convincingly demonstrate the difference between background and real events. Using a PE isotype control AND an FMO, the authors can properly assess background staining (inherent to intracellular staining) and then calculate how many PILTs (not just %, but absolute numbers) are present in WT, CD1d^{-/-} and b2m^{-/-} mice.

These two simple but critically important experiments will ensure that there won't be doubts in the field if PILTs are real or an experimental artifact of transgenic mice. This will also be needed to avoid confusion between the conclusions of the initial Nat Comm paper and the current submission. The RNAseq data that are included are not sufficient to do this since transcript expression does not mean that protein is expressed. However, the flow experiments outlined above will address this.

Reviewer #3

(Remarks to the Author)

The authors have addressed all of my concerns. Their responses were thorough and clarified all the points I raised. The addition of the human dataset comparison, in my opinion, also nicely adds to the study. I have no further questions or issues.

Version 2:

Reviewer comments:

Reviewer #2

(Remarks to the Author)

The authors now fully address all of my previous comments and now provide compelling evidence for the expression of MHC class I on DPs. This will be immensely helpful down the road and prevent a prolonged discussion whether PILTs are real or an experimental artifact.

Both figures from their letter need to be included - (Fig. 1 from the letter is Suppl Fig9a, but it's unclear where Fig. 2 is shown). Particularly the PILT #s in WT, CD1d^{-/-} and b2M^{-/-} mice are an important data piece for the field.

NCOMMS-24-74534

Response to review:

We thank all three reviewers for their constructive comments.

Reviewer #1 (Remarks to the Author):

This study presented by Hassan et al. explores the development of peptide-specific PLZF+ innate-like T (PILT) cells using single-cell transcriptomic analysis. The researchers found that PILT cells share transcriptional profiles with invariant Natural Killer T (iNKT) cells but have a polyclonal TCR repertoire, suggesting MHC I restriction and broader antigen specificity. They also showed that artificial thymic organoid cultures (ATOC) support PILT cell development and that TCR specificity influences their lineage commitment and differentiation.

This paper presents a comprehensive collection of data and analyses that provide valuable insights into the transcriptional profile of a distinct subset of innate-like T cells. These findings are well-positioned to enhance the field's understanding of innate-like T cells and how their subsets compare and contrast. However, there remain some technical questions that require further clarification, and a recommendation that this body of work may belong in a setting that is more appropriate for the amount and type of data that is provided, such as a Resource paper in an immunology journal.

Critiques:

- It is unclear whether cells were pooled for the scRNAseq experiments described in Extended Fig. 1b, or how this process was carried out. Additional details are needed in the narrative, or a clearer diagram should be provided to explain how this experimental design resulted in the UMAPs shown in Fig. 1.

Answer: We thank the reviewer for this comment. In the revised manuscript, we provide a comment in the text on **page 5 line 118-119** clarifying how the experimental design resulted in the UMAPs shown in Fig. 1. In addition, we included detailed information for each single-cell RNA sequencing experiment in the provided Source Data file.

- Authors do not provide any absolute numbers that describe the cell populations of interest, only percentages, which does not create a complete picture of what might be happening with other cell types.

o For example, in Figure 5c and d, authors are making a claim that in this AOTC model there is an emergence of PILT cells and iNKT cells, but do not provide the absolute cell numbers over time. There could be changes in other cell populations that influence the percentage of PILT cells without the PILT cell population actually increasing.

Answer: The reviewer's point is well taken. In the current version, we add a comment on this in the text on **page 11 lines 313-316** and these data are incorporated in **Figure 5c** and **5e**.

- There is no discussion of what the function of these cells might be and why they might be important to study. How would these cells respond to activation in vitro? What cytokines are they capable of producing? Is there any evidence that suggests that their function may differ from other innate-like T cells function given that they are transcriptionally similar?

Answer: We agree with the reviewer that we should provide further discussion on the function of PILT cells. In the revised manuscript, we included two additional paragraphs on **page 16 lines 445-466** drawing a parallel between PILT and iNKT cells and discussing on possible differences regarding function.

- As mentioned in the text of this paper, this group has already published about the identification of the PILT cell subset and this paper seems to be building off of the previous publication, characterizing its transcriptional profile, without actually concluding anything novel about these cells. Because of this, this work seems to lend itself better as a resource rather than as a standalone research article.

Answer: We regret this assessment and wish to emphasize that in this study in addition to characterizing the PILT-cell transcriptional profile at single cell level, we pinpoint the transcriptional and developmental similarities between PILT cells and other members of the innate-like T cell lineage. Moreover, we show that PILT cells exhibit a "truly" polyclonal TCR repertoire with similar characteristics to conventional CD8 T-cell TCR repertoire. Furthermore, by establishing a novel "on-time" T-cell receptor retrogenic ATOC system, we were able to reproduce thymic PILT cell development in *in vitro* settings and provide evidence for an instructive role of TCR specificity in PILT cell lineage commitment and functional differentiation. To our knowledge, these are all novel

findings building on our current understanding of innate-like T cell development in the thymus.

Considering these information, we hope that the reviewer will agree that the present study exhibit adequate novelty for a standalone research article.

Reviewer #2 (Remarks to the Author):

In this manuscript, Georgiev et al. examine PLZF+ MHC class I-restricted T cells (PILTs) in the thymus.

PLZF expression in the thymus occurs in NKT cells and other innate-like T cells such as MAIT cells. A key difference in thymic selection between conventional T cells and NKT cells is that conventional T cells are selected by thymic epithelial cells, while NKT cells get selected by adjacent double positive (DP) thymocytes. This occurs, because DP thymocytes in WT mice express CD1d, but not conventional MHC class I or II. In a previous study, the authors generated mice that forced expression of MHC class I on DPs in CD1d deficient mice and observed that this led to the generation of MHC class I-restricted PLZF+ T cells (PILTs). In this manuscript, the authors further examine these PILTs using scRNAseq (on ex vivo isolated cells) and thymic organoid cultures.

Overall, the experiments are well designed and well described to build a very coherent and thorough body of work. My main question is in regards to the conclusions and the biological relevance of these PILTs. The authors seem to suggest that PILTs are a naturally occurring T cell lineage (the last sentence in the discussion even suggests that these cells occur in humans).

However, generation of these cells requires the experimentally forced expression of MHC class I on DPs (T MHC I mice). I did not see any compelling evidence in this manuscript or the initial report indicating that PILTs develop in WT mice. Since WT DPs do not express MHC class I, how could PILTs even develop in WT animals?

Answer: The reviewer's point is well taken. Indeed, in the initial report (Georgiev et al., 2021) we showed that MHC class I expression is downregulated at the DP stage of thymocyte development in WT mice. Notably, this was only assessed with the use of flow cytometry and bulk RNAseq analysis of total DP thymocytes (bulk RNAseq data

was obtained from ImmGen). Yet, by re-analyzing a publically available scRNAseq dataset (B6 WT thymocytes (Steier *et al.*, 2023)), now we are able to show that in B6 WT mice there is a sizable fraction of MHC class I expressing cells (*H2-K1* or *H2-D1*) within each of the DP-cell populations described in the referred study (data provided below in **Figure 1a for the reviewers**). Moreover, these MHC I expressing cells co-express *Slamf6* (data provided below in **Figure 1b for the reviewers**) and *Slamf1* (data not shown) rendering them capable of selecting PILT cells. Of note, human thymocytes exhibit a similar expression pattern of HLA molecules shown as data provided below in **Figure 1c for the reviewers** (data extracted from an atlas of human thymic development scRNAseq dataset (Park *et al.*, Science, 2020)).

Although, it is not clear why some DP thymocytes upregulate MHC I expression at this stage, in the revised manuscript we provide additional data which is suggestive of an interferon signaling signature in some MHC I expressing DP thymocytes (see in **Supplementary Figure. 9b**). Therefore, steady-state production of interferons in the thymus might mediate MHC I upregulation on some thymocytes during the DP stage of their development.

In addition, in our initial report (Georgiev *et al.*, 2021), we already showed that a small fraction of PILT cells develop in WT mice, which do not require CD1d expression for their development in the thymus (Georgiev *et al.*, 2021). Now, in **Figure 1d for the reviewers** (right panel), we provide further evidence that PILT-cell development is abrogated in *B2m^{-/-}* animals.

Taken together, these data infers that:

- 1) In B6 WT mice, there is a sizable fraction of DP thymocytes co-expressing MHC I, *Slamf1* and *Slamf6*, rendering them capable of selecting PILT cells in WT settings.
- 2) PILT cell development is not CD1d dependent, yet requires *B2m* expression in WT mice which further infers that a small fraction of PILT cells develop as MHC-I restricted cells in WT animals.

In the revised manuscript, **we included a new Supplementary Figure. 9a** displaying the base-level expression of MHC I on DP thymocytes in WT animals and **a new Supplementary Figure. 9b** showing co-expression of MHC I and IFN-stimulated genes

(ISGs) such as *Isg15*, *Isg20* and *Ifi203* on a fraction of DP thymocytes. In addition, we included a new paragraph in the discussion section **page 16 lines 460-466** discussing the possible impact of sterile interferon production in the thymus on PILT cell development and function.

Figure. 1 | MHC-I expression levels on WT DP thymocytes

a. Violin plots showing *H2-K1*, *H2-D1* and *Cd1d1* normalized expression levels within DP (P), double positive proliferating; DP (Q1), DP quiescent 1; DP (Q2), DP quiescent 2; DP (Sig.), DP signaled; Mature CD4 and Mature CD8 thymocyte-cell populations as annotated by (Steier *et al.*, Nat. Immunol, 2023). **b.** Feature scatter plots showing *H2-K1*, *H2-D1* and *Slamf6* co-expression levels within thymocyte-cell populations as annotated by (Steier *et al.*, Nat. Immunol, 2023). Data shown in **a** and **b** were obtained from B6 WT scRNAseq dataset described in Steier *et al.*, 2023. **c.** Violin plots showing *HLA-A*, *HLA-B*, *HLA-C* and *CD1D* normalized expression levels within different human thymocyte-cell populations as annotated by (Park *et al.*, Science, 2020). Data shown in **c** were obtained from human scRNAseq dataset described in Park *et al.*, Science, 2020. **d.** Representative flow cytometry plots (left) and a scatter plot (right) showing thymic PILT-cell frequencies in B6 WT, B6 *CD1d*^{-/-} and B6 *B2m*^{-/-} mice. PILT cells are gated as Live/CD19⁻CD1d⁻Tet⁺MR1⁻Tet⁺TCRb⁺PLZF⁺. Data are representative of three independent experiments in **d**. Unpaired two-tailed Student's *t*-test was performed in **d**. Data are presented as mean values ± SD.

The intriguing findings to me were related to the PILT – NKT comparisons made by the authors, which do provide very unique and novel insight to the potential role of CD1d/antigen vs. DP-mediated selection in the functional development of NKT cells.

In its current state the manuscript is rather confusing, because the key conclusion is not really clear. If the authors truly want to argue that PILTs occur in a WT environment (and the T MHC I mice are merely a tool to increase PILT numbers to enable scRNAseq, etc. experiments) then additional experiments are required to support this notion. If the intended main focus is on using PILTs to improve our understanding of NKT cell development, then editing for clarification is sufficient. Finally, I do want to emphasize again that this is a very compelling, creative and intriguing set of experiments that provides unique insight into thymic selection and cell fate decision processes.

Answer: As we mentioned above, in our initial report, we already showed that a small fraction of PILT cells develop in WT mice. These cells were found to be CD1d independent and here we further show (as data for the reviewer) that PILT-cell development is abrogated in a B2m^{-/-} mice. In addition, we provided a set of evidences that in WT mice, a prominent fraction of DP thymocytes co-express classical MHC I, *Slamf6* and *Slamf1* rendering them capable of selecting PILT cells in WT settings. In the present study, we utilized the CD1d^{-/-} mouse strain as a suitable model for studying WT PILT-cell development due to complete abrogation of NKT-cell development and therefore avoiding possible contamination of sorted PILT cell-enriched fractions with CD1d restricted NKT cells with diverse TCRs. In the revised version of the manuscript, we included a comment in the text on **page 5 lines 112-115** clarifying why the CD1d^{-/-} mouse strain is a suitable model for studying WT PILT-cell development.

We hope that with the provided new data, and additional discussion, we improved the overall clarity of the manuscript and resolved the confusions regarding the key conclusions of the manuscript.

Lastly, we would like to thank the reviewer for the positive feedback.

Reviewer #3 (Remarks to the Author):

In this study, Hassan et. al., nicely describes the transcriptional profiles and potential developmental features of antigen-specific PLZF+ innate-like T cells compared with iNKT cells, utilizing multi-modal single cell sequencing (RNA + TCR) as well as an in vitro thymic organoid culture system. This paper is well-written and provides unique insight into this newly described T cell type, its potential broad functionality and its development. However, the study has some gaps that should be addressed to clarify the conclusions and strengthen the overall impact of the findings.

Main comments:

1. Based on the lack of unique surface markers to identify PILT cells, authors used a simple strategy to enrich more PILT cells. Using Zbtb16 (PLZF gene) expression as a marker for this population in their scRNAseq clustering, they found a wide range in Zbtb16 expression and, what looks like an overall lower gene expression in PILT-1 cells, across experiments. At a protein level, there also seems to be a range of expression in the PILT population. It would be easy to confirm that PLZF-low PILTs are transcriptionally bias towards PILT-1 type subsets and would overall help the clarity of the study; and that these populations are really the expected PILTs.

Answer: Indeed the PLZF^{low} PILT cells are biased toward PILT1 phenotype, this is true on both protein and mRNA levels (see the figure below). In addition, we have displayed the Zbtb16 mRNA expression levels across PILT cell subsets in **Figure 2b** (in the manuscript) showing that PILT1 cells in clusters 3-6 exhibit the lowest Zbtb16 expression levels.

2. It is somewhat surprising that PILT and iNKT cells completely overlap in Fig 3. How were these umaps harmonized? Are there unique RNA features that distinguish this population from iNKT beyond TCR expression?

Answer: PILT and iNKT cells were integrated using SCVI with $n_{\text{latent}}=30$. Moreover, the same holds when using other integration methods such as Harmony and CCA (see the figure below). Looking at the differentially expressed genes between corresponding PILT and iNKT cells we could not identify unique features or cell markers distinguishing these two populations. We see some expression differences in genes related MHC I loading and processing which can be attributed to the forced upregulation of Nlrc5 (a transcriptional regulator of these genes).

3. The distribution of the PILT cell states (1, 2, 17 etc) is different between WT and T-MHC mice. (Fig 3 c, 2nd plot) This is a very interesting finding and may provide some insight into alterations in PILT development. Authors should discuss this in more detail.

Answer: Indeed, this is an interesting question. In the revised manuscript, we included further discussion and provided two non-mutually exclusive hypotheses regarding this finding in the text **on page 15 line 420-430**.

4. Some of the most interesting findings were those in the transgenic ATOC system, however authors have limited discussions on these findings. Of particular interest are: 1) There is marked loss of PILT cells in the ATOC system at 5 weeks. (Fig 5) Why do these cells develop and disappear? It is technical or biological? 2) Only two out of the three PILT1 TCR clones exhibit PLZF expression. (Fig 8) Why might this be?

Answer 1): The reviewer's point is well taken and we apologize that we did not discuss on this in the initial submission. By week 5 the ATOC system becomes overgrown by the MS5 stromal cells and reaches a point where the supplemented media is no longer sufficient to provide a suitable level of nutrient/cytokines required to sustain the system resulting in dramatic decrease in cell viability. Therefore, the observed disappearance of these cells is of rather technical than biological reasons. We included this information in the text **on page 11 line 313-316** and these data are incorporated in **Figure 5e**.

Answer 2): We can only speculate on the reason why these two clones develop as innate-like T cells while the third one does not. For example, on the one hand, the third clonotype might be selected on an antigen, which is expressed only by DP thymocytes developing *in vivo* in the thymus, but not by ATOC derived DP thymocytes. On the other hand, this clonotype might recognize antigen which is abundantly expressed by the MS-5 stromal cell line, lowering its chance to be selected by neighboring DP thymocytes.

5. On a broader level, age plays an important role in the innate-like properties of T cells (Reynaldi, PNAS, 2019; Smith, Cell, 2018). 1) Does the mouse age used in these studies, or the thymic stromal cell culture "age" play a role in the frequency or "state" (1, 2, 17 etc) of PILT? 2) What about other basic features such as sex or mouse strain?

Answer 1): We agree that, similar to iNKT cells, age might influence the frequency and "state" of PILT cells. Therefore, in all scRNAseq experiments we used only 10 weeks old animals in order to minimize such effects. For the *in vitro* ATOC experiments, all donor mice were 8-12 week old. We already addressed the stromal cell culture "age" question in the previous comment (see above comment 4)

Answer 2): We used both female and male animals in all experiments and we did not observe any differences related to PILT-cell development between both sexes. However, we have previously shown that on mixed B6/BALBc F1 background, PILT cells

expand in similar fashion to iNKT cells with more prominent PILT2 population (Fig. 6 Georgiev et al, Nat Comm, 2021).

6. How do these new innate-like T cells in mouse related to the more innate-like populations recently identified in human blood? (Thomson, Nat Imm, 2023; Terekhova, Immunity, 2023). Human thymic datasets could be used to compare overall transcriptional similarities between mouse and human subsets (Park, Science, 2020).

Answer: We thank the reviewer for raising this interesting suggestion. Indeed, when integrating our murine PILT dataset with the suggested human thymic dataset (Park, Science, 2020), murine PILT (mPILT) cells clustered next to human NKT-like cells and other unconventional populations with innate-like phenotype such as CD8aa, a fraction of gd T cells and a PLZF+ thymocyte population referred to by *Park et al.* as Th17-like cells. We included this information in the text on **page 10 lines 264-267** and these data are incorporated in **Figure 3h**.

Unfortunately, we could not reliably integrate PILT cells with MNP-1 or MNP-2 cells from Thomson, Nat Imm, 2023, as the published data did not annotate either of the cells (see below for the available annotation). The data from Terekhova, Immunity, 2023, was annotated correctly; however, the available data contained only the normalised expression matrix but not the raw count, which SCVI cannot use for integration.

```
> table(GEX$seurat_pbmc_type)
```

B cell progenitor	CD14+ Monocytes	CD16+ Monocytes	CD4 Memory	CD4 Naive	CD8 effector	CD8 Naive	Dendritic cell	Double negative T cell
271	311	68	241017	19366	19288	24812	23	21558
NK cell	pDC	pre-B cell						
25634	3	833						

```
> |
```

Minor comments:

1. The distribution umap plots in Figure 3 are hard to understand as presented, especially 3b, 3g and 3i.

Answer: In **Figure 3b, 3g, and 3i** (in the revised manuscript version corresponding to **Figure 3b, 3e, and 3g**, respectively), we present data in two ways:

- 1) Left panels showing UMAP overlay of iNKT and PILT cells in **3b** and **3e** or MAIT and PILT cells in **3g**.
- 2) Right panels showing the proportion evaluation of PILT to iNKT (in **3b** and **3e**) and PILT to MAIT cells (in **3g**) in each of the clusters as defined in **Figure 3a, 3d** and **3f**,

respectively. Therefore, displaying an important information regarding cell type distribution across the different clusters.

In the revised version of the manuscript, we altered the text in the Figure 3 legend for a better clarity.

2. Experiments were ran across a number of different sequencing runs, which can impact data quality (# of cells per samples, read counts per cells etc). Are there any batch effects in these data? It would be good to provide sample QC metrics for these datasets.

Answer: Using SCVI we were able to mitigate the batch effect between the different runs. We provide a link to the summary file output from the cellranger showing all relevant metrics for all of our single-cell runs.

https://drive.google.com/drive/folders/1-4CUD6g70k0ZFgj0cTaBzyYZpOK_1Y9

3. The coloration used for gene expression is not consistent throughout the manuscript. It would be helpful to keep this consistent (e.g. black is always low, white is always high).

Answer: We thank the reviewer for raising this question. In the current version of the manuscript, we adapted the coloration of all density plots (Fig. 1b, 6c, 7d and Extended Data Fig. 2c-f) so that all gene expression data is displayed as: black is always high and white is always low.

4. Fig 5c, 8e, 8g: There are no basic stats on these graphs. P-values should be included.

Answer: In the current version, we included statistics on these graphs.

5. Fig 8 c – cannot tell the colors of the clone in the umap.

Answer: We adjusted the colors of the highlighted clonotypes and included numbering so that each clonotype is clearly visible on the UMAP.

Response to review:

Reviewer #1 (Remarks to the Author):

The authors have provided new data and revised the manuscript. The revised version has adequately addressed the concerns raised about the original version.

We are pleased that our response satisfied the reviewer.

Reviewer #2 (Remarks to the Author):

I stated in my initial review that “If the authors truly want to argue that PILTs occur in a WT environment (and the T MHC I mice are merely a tool to increase PILT numbers to enable scRNAseq, etc. experiments) then additional experiments are required to support this notion.”

The authors make it clear in the response that they indeed argue PILTs occur in a WT environment. I am somewhat surprised by this since in the initial report Georgiev and Hogquist concluded (last sentence of the abstract): “We demonstrate that PIL and NKT cells compete for a narrow niche, suggesting that the absence of peptide-MHC on DP thymocytes facilitates selection of non-peptide specific lymphocytes.”

(1) Now, the authors are stating that peptide-MHC are in fact NOT absent on DP thymocytes. This is an important distinction from the initial report and a very different conclusion. There are 2 bare minimum experiments that they need to do to support this new conclusion:

1. (2) Use flow cytometry to show which DPs express MHC class I (in context of CD1d expression) with proper staining controls (FMO). (3) The expression pattern & levels in WT vs T MHC I mice should be compared and discussed as they are pertinent for the observed skewing in PILT 1 vs 17.

Answer (1): Indeed, in the initial report (Georgiev et al., 2021) we used flow cytometry analysis to show that MHC class I expression is downregulated at the DP stage of thymocyte development in WT mice. However, we only compared MHC I expression on DP thymocytes to CD4 SP and CD8 SP thymocytes without the use of isotype control

and therefore we were not able to assess any remaining low-level expression of MHC-I on DP thymocytes (Georgiev et al., 2021, Figure 1e). Now, as requested, we provide these stainings with the adequate controls (data provided below in **Figure. 1 for the reviewers**).

Answer (2): We agree with the reviewer's comment that "*transcript expression does not mean that protein is expressed*" and we thank the reviewer for the suggested experiments.

Now we provide a flow cytometry evaluation of MHC class I expression on DP thymocytes from WT, CD1d^{-/-} and T-MHC I transgenic mice, including isotype and FMO controls. Moreover, we included thymocytes from B2m^{-/-} mice (deficient of MHC I surface expression) as an additional negative control in this staining, which serves as an arguably better control for background staining with this antibody (data provided below in **Figure. 1 for the reviewers**).

Figure. 1 | Flow cytometry evaluation of MHC I expression on DP thymocytes.

Representative flow cytometry plots (left) and a scatter plot (right) showing MHC I expression on thymocytes from WT, CD1d^{-/-}, T-MHC I and B2m^{-/-} transgenic mice. Each point represents one animal: n = 4 animals per group. Data are representative of two independent experiments. One-way ANOVA followed by Fisher's LSD multiple comparisons test was performed; ns, not significant ($p \geq 0.05$), *** $p < 0.001$, and **** $p < 0.0001$. Data are presented as mean values \pm SD.

As shown in **Figure. 1 for the reviewers**, WT DP thymocytes maintain a low base-expression level of MHC-I (albeit at lower levels than T-MHC DP thymocytes) facilitating PILT-cell selection in WT settings. Importantly, DP thymocytes from WT and CD1d

deficient mice exhibit comparable MHC I expression levels suggesting that the lack of CD1d expression does not alter MHC I expression level on DP thymocytes in the CD1d^{-/-} mouse model used in our study.

We are happy to include this data in the manuscript, as **a new Supplementary Figure. 9a** displaying flow cytometry evaluation of MHC I expression on DP thymocytes in WT, CD1d^{-/-} and T-MHC I transgenic mice.

Answer (3): This is already included in the discussion section “*On the one hand, forced expression of MHC I molecules on DP thymocytes may favor development of PILT1 and PILT2-cell subsets and to a lesser extent PILT17 cells. On the other hand, the reduced PILT17 frequency in the MHC-PILT cell pool might be a result of a competition for a limited cellular niche supporting PILT17-cell development in the thymus.*” **page 15 line 427-430.**

2. In Figure 1d of their response letter, the authors still see a few PILT cells in b2m^{-/-} mice. I assume the authors would argue that those events are “background”. I would agree with this in principle (PLZF antibody that didn’t get washed out after the intracellular stain), but since the population is so rare, it is essential to convincingly demonstrate the difference between background and real events. Using a PE isotype control AND an FMO, the authors can properly assess background staining (inherent to intracellular staining) and then calculate how many PILTs (not just %, but absolute numbers) are present in WT, CD1d^{-/-} and b2m^{-/-} mice.

Answer: The reviewer’s point is well taken. As suggested, we repeated the staining including an isotype and an FMO controls (data provided below in **Figure 2 for the reviewers**). In addition, we included an anti-F4/80 antibody in our staining panel in order to exclude macrophages as a possible source of background in our analysis.

In line with what we have shown before (Georgiev et al., 2021, Figure 3b right panel), our results show that PILT cells in CD1d^{-/-} are slightly increased in frequency and numbers in comparison to WT mice. Notably, the total numbers of cells in the PILT-cell gate are approximately eight fold reduced in thymi from B2m^{-/-} mice in comparison to WT mice inferring PILT-cell development abrogation in B2m^{-/-} mice (see below **Figure 2 for the reviewers**).

Of note, although an isotype control provides a good estimate for a staining background of an antibody, the true background of the anti-PLZF antibody used in our study can be only assessed with the use of PLZF^{-/-} mouse as negative control.

Figure. 2 | Flow cytometry evaluation of PILT-cell frequencies and numbers in the thymus.

Representative flow cytometry plots (left) and a scatter plot (right) showing thymic PILT-cell frequency (plotted on the left axis) and number (plotted on the right axis) from WT, CD1d^{-/-}, and B2m^{-/-} transgenic mice. Values are calculated by subtracting Isotype controls from the PLZF staining. PILT cells are gated as Live/CD19⁺F4/80⁺CD1d⁻Tet⁻MR1⁻Tet⁻TCRb⁺PLZF⁺. Each point represents one animal: n = 4 animals per group. Data are representative of two independent experiments. One-way ANOVA followed by Fisher's LSD multiple comparisons test was performed; ns, not significant ($p \geq 0.05$), * $p < 0.05$, and **** $p < 0.0001$. Data are presented as mean values \pm SD.

These two simple but critically important experiments will ensure that there won't be doubts in the field if PILTs are real or an experimental artifact of transgenic mice. This will also be needed to avoid confusion between the conclusions of the initial Nat Comm paper and the current submission. The RNAseq data that are included are not sufficient to do this since transcript expression does not mean that protein is expressed. However, the flow experiments outlined above will address this.

Answer: We hope that the provided new data will dispel remaining doubts regarding PILTs existence in WT mice.

Reviewer #3 (Remarks to the Author):

The authors have addressed all of my concerns. Their responses were thorough and clarified all the points I raised. The addition of the human dataset comparison, in my opinion, also nicely adds to the study. I have no further questions or issues.

We are pleased that our response satisfied the reviewer.

Response to review:

Reviewer #2 (*Remarks to the Author*):

The authors now fully address all of my previous comments and now provide compelling evidence for the expression of MHC class I on DPs. This will be immensely helpful down the road and prevent a prolonged discussion whether PILTs are real or an experimental artifact.

We are pleased that our response satisfied the reviewer.

Both figures from their letter need to be included - (Fig. 1 from the letter is Suppl Fig9a, but it's unclear where Fig. 2 is shown). Particularly the PILT #s in WT, CD1d^{-/-} and b2M^{-/-} mice are an important data piece for the field.

Answer: We are happy to include this data in the manuscript, as **a new Supplementary Figure. 5**